# ZERO-SHOT OUT-OF-DISTRIBUTION DETECTION WITH FEATURE CORRELATIONS

## ABSTRACT

When presented with Out-of-Distribution (OOD) examples, deep neural networks yield confident, incorrect predictions. Detecting OOD examples is challenging, and the potential risks are high. In this paper, we propose to detect OOD examples by identifying inconsistencies between activity patterns and class predicted. We find that characterizing activity patterns by feature correlations and identifying anomalies in pairwise feature correlation values can yield high OOD detection rates for far-from-distribution examples. We identify anomalies in the pairwise feature correlations by simply comparing each pairwise correlation value with its respective range observed over the training data. Unlike many approaches, this can be used with any pre-trained softmax classifier and does not require access to OOD data for fine-tuning hyperparameters, nor does it require OOD access for inferring parameters. The method is applicable across a variety of architectures and vision datasets and, for the important and surprisingly hard task of detecting far-from-distribution outliers, it generally performs better than or equal to state-of-the-art OOD detection methods (including those that do assume access to OOD examples).[1]

## 1 INTRODUCTION

Even when deep neural networks (DNNs) achieve impressive accuracy on challenging tasks, they do not always visibly falter on misclassified examples: in those cases they can often make predictions that are both very confident and completely incorrect. Yet, predictive uncertainty is essential in real-world contexts tolerating minimal error margins such as autonomous vehicle control and medical, financial and legal fields.

In this work, we focus on flagging test examples that do not contain any of the classes modeled in the train distribution. Such examples are often referred to as being *out-of-distribution* (OOD), and while their existence has been well-known for some time, the challenges of identifying them and a baseline method to do so in a variety of tasks such as image classification, text classification, and speech recognition were presented by Hendrycks and Gimpel (2017). Recently, Nalisnick et al. (2019a) identified a similar problem with generative models: they demonstrate that flow-based models, VAEs, and PixelCNNs cannot distinguish images of common objects such as dogs, trucks, and horses (i.e. CIFAR-10) from those of house numbers (i.e. SVHN), assigning a higher likelihood to the latter when the model is trained on the former. They report similar findings across several other pairs of popular image datasets.

While we might expect neural networks to respond differently to OOD examples than to in-distribution (ID) examples, exactly where and how to find these differences in activity patterns is not at all clear. Hendrycks and Gimpel (2017) and others (Nguyen et al., 2015; Yu et al., 2011) showed that looking at the maximal softmax value is insufficient. In Section 2 we describe some other recent approaches to this problem. In this work, we find that characterizing activity patterns by feature correlations—computed with an extension of Gram matrices that we introduce—lets us quantify anomalies to allow state-of-the-art (SOTA) detection rates on OOD examples.

**Intuition.** We identify out-of-distribution examples by jointly considering the class assigned at the output layer and the activity patterns in the intermediate layers. For example, if an image is

---

[1]The code for this work is available at `https://github.com/zeroshot-ood/ood-detection`

predicted to be a dog, yet the intermediate activity patterns are somehow atypical of those seen by the network for other dog images during training, then that is a strong indicator of an OOD example. This effectively allows us to detect incongruence between *the prediction made by the network* and *the path by which it arrived at that prediction*.

**Strengths.** Unlike those previous works that assume access to OOD examples and train an auxiliary classifier for identifying anomalous activity patterns, our method finds differences in activity patterns without requiring access to any OOD examples, and it works across architectures. We hope this will also help further our understanding of how neural networks respond differently to OOD examples *in general*, not just how a particular network responds to examples coming from a particular distribution.

**Contributions.** This work includes the following contributions:

1. We extend Gram matrices to compute effective feature correlations.
2. Using the $p^{th}$-order Gram matrices, we present a new technique for computing class-conditional anomalies in activity patterns.
3. We evaluate this technique on OOD detection, testing on
   - competitive architectures: DenseNet, ResNet;
   - benchmark OOD datasets including: CIFAR-10, CIFAR-100, SVHN, TinyImageNet, LSUN and iSUN.
4. Zero-shot: crucially, *our method does not require access to OOD samples* for tuning hyperparameters or for training auxiliary models.
5. Nevertheless, we report results which, for the challenging and important cases of far-from-distribution examples, are generally better than or equal to the state-of-the-art method for OOD detection that does require access to OOD examples.

## 2 RELATED WORK

Previous work which aims to improve OOD detection can be roughly grouped by several themes:

**Bayesian Neural Networks.** A nice early Bayesian approach (Gal and Ghahramani, 2016) estimates predictive uncertainty by using an ensemble of sub-networks instantiated by applying dropout at test time. As opposed to implicitly learning a distribution over the predictions by learning a distribution over the weights, Chen et al. (2019) and Malinin and Gales (2018) explicitly parameterize a Dirichlet distribution over the output class distributions using DNNs in order to obtain a better estimate of predictive uncertainty; the main differences between these methods is that Chen et al. (2019) use ELBO, which only requires the in-distribution dataset for training whereas Malinin and Gales (2018) use a contrastive loss which requires access to (optionally synthetic) OOD examples.

**Using any pre-trained softmax deep neural network with OOD examples.** Lee et al. (2018b)—to the best of our knowledge, the current SOTA technique by a significant margin—compute the Mahalanobis distance between the test sample's feature representations and the class-conditional gaussian distribution at each layer; they then represent each sample as a vector of the Mahalanobis distances, and finally train a logistic regression detector on these representations to identify OOD examples. Another technique in this category is ODIN (Liang et al., 2018): they use a mix of temperature scaling at the softmax layer and input perturbations to achieve better results. In fact, both Lee et al. (2018b) and Liang et al. (2018) add small input perturbations to achieve better results; the former do so to increase the confidence score, while the latter do so to increase the softmax score. Recently, Quintanilha et al. (2019) achieve results comparable to that of Lee et al. (2018b) by training a logistic regression detector that looks at the means and standard deviations of various channels activations. Unlike the previous two techniques, Quintanilha et al. (2019) achieves comparable results even without the use of input perturbations, which allows it to be applicable to non-continuous domains. Our work, too, does not involve input perturbations.

All of these techniques depend on OOD examples for fine-tuning hyperparameters (Liang et al., 2018) or for training auxiliary OOD classifiers (Lee et al. (2018b); Quintanilha et al. (2019)). Furthermore, these classifiers neither transfer between one non-training distribution and another, nor do they transfer between networks, so separate classifiers must be trained for each (In-Distribution, OOD,

Architecture) triplet. In many real-world applications, this is infeasible: we cannot assume advance access to all possible OOD distributions. Our work does not require access to OOD samples.

**Alternative Training Strategies.**   Lee et al. (2018a) jointly train a classifier, a generator and an adversarial discriminator such that the classifier produces a more uniform distribution on the boundary examples generated by the generator; they use OOD examples to fine-tune hyperparameters. DeVries and Taylor (2018) train neural networks with a multi-task loss for jointly learning to classify and estimate confidence. Shalev et al. (2018) use multiple semantic dense representations as the target instead of sparse one-hot vectors and use a cosine-similarity based measure for detecting OODs. Building on the idea proposed by Lee et al. (2018a), Hendrycks et al. (2019a) propose an *Outlier Exposure* (OE) technique. They regularize a softmax classifier to predict uniform distribution on (any) OOD distribution and show the resulting model can identify examples from unseen OOD distributions; this differs significantly from previous works which used the same OOD distributions for both training and testing. Unlike other methods, they retain the architecture of the classifier and introduce just one additional hyperparameter—the regularization rate—and also demonstrate that their model is quite robust to the choice of OOD examples chosen for the regularization. However, while the OE method is able to generalize across different non-training distributions, it understandably does not achieve the rates of Lee et al. (2018b) on most cases. Recently, Hendrycks et al. (2019b) make significant advances in detecting near-distribution outliers without having any knowledge of the exact out-of-distribution examples by using in-distribution examples in a self-supervised training setting.

**Generative Models.**   Ren et al. (2019) hypothesize that stylistic factors might impact the likelihood assignment and propose to detect OOD examples by computing a likelihood ratio which depends on the semantic factors that remain after the dominant stylistic factors are cancelled out. On the other hand, Nalisnick et al. (2019b) argue that samples generated by a generative model reside in the typical set, which might not necessarily coincide with areas of high density. They demonstrate empirically that OOD examples can be identified by checking if an input resides in the typical set of the generative model. Unlike the standard experimental setting, they aim to identify distributional shift, which predicts if a batch of examples are OOD.

## 3   EXTENDING GRAM MATRICES FOR OUT-OF-DISTRIBUTION DETECTION

**Overview**   In light of the above considerations, we are interested in proposing a method that does not require access to any OOD examples, that does not introduce hyperparameters that need tuning, and that works across architectures. Gram matrices can be used to compute pairwise feature correlations, and are often used in DNNs to encode stylistic attributes like textures and patterns (Gatys et al., 2016). We extend these matrices as will be described below, and then use them to compute class-conditional bounds of pairwise feature correlations at multiple layers of the network. Starting with a pre-trained network, we compute these bounds over only the training set, and can then use them to effectively discriminate between in-distribution samples and out-of-distribution samples at test time. Unlike other SOTA algorithms, we do not need to "look" at any out-of-distribution samples to tune any parameters; the only tuning required is that of a normalizing factor, which we compute using a randomly-selected validation partition of the (in-distribution) test set.

**Notation**   If the considered deep convolutional network has $L$ layers and the $l^{th}$ layer has $n_l$ channels, we consider feature co-occurrences between the $\sum_{1<=l<=L} \frac{n_l*(n_l+1)}{2}$ pairs of feature-maps. (Note that by "layer" we refer to any set of values obtained immediately after applying convolution or activation functions.) We use the following notation:

| | |
|---|---|
| $F_l(D)$ | The feature map at the $l$-th layer for input image $D$; when referring to an arbitrary image $D$, we just write $F_l$. It can be stored in a matrix of dimensions $n_l \times p_l$, where $n_l$ is the number of channels at the $l$-th layer and $p_l$, the number of pixels per channel, is the height times the width of the feature map. |
| $D_c/f(D)$ | The predicted class for input image $D$ |
| *Train* | The set of all train examples |
| Va | The set of all validation examples. 10% of the examples not used in training are randomly chosen as validation examples. |
| Te | The set of all test examples, disjoint as usual from the training and validation sets. We assume that only the test set may contain out-of-distribution examples. |

**Gram Matrices and Higher order Gram Matrices**    We compute pairwise feature correlations between channels of the $l$-th layer using the Gram matrix:

$$G_l = F_l F_l^\top \tag{1}$$

where $F_l$ is an $n_l \times p_l$ matrix as defined above.

In order to compute feature correlations with more prominent activations of the feature maps, we define a *higher-order gram matrix*, which we write $G_l^p$, to be a matrix computed identically to the regular Gram matrix, but where, instead of using a raw channel activation $a$, we use $a^p$, the $p^{th}$ power of each activation. $G_l^p$ is therefore computed using $F_l^p$, where the power of $F_l$ is computed element-wise; in an effort to retain uniform scale across all orders of Gram matrices for a given layer, we compute the (element-wise) $p$-th root. The $p$-th order gram matrix is thus computed as:

$$G_l^p = \left( F_l^p F_l^{p\top} \right)^{\frac{1}{p}} \tag{2}$$

We show in Section 5 that higher $p$ values help significantly in improving the OOD detectability. In our experiments, we limit the value of $p$ to 10, as exponents beyond 10 are not worth the extra computation that is needed to avoid overflow errors[2].

The flattened upper (or lower) triangular matrix along with the diagonal entries is denoted as $\overline{G_l^p}$. The set of all orders of gram matrices (in our case $\{1, \ldots 10\}$) to be considered is denoted by $P$. The schematic diagram of the proposed algorithm is shown in Fig. 15 (in Appendix A).

**Preprocessing**    If we compute $\overline{G_l^p}$ for every layer $l$ and every order $p \in P$, we obtain a total of $N_S = \sum_{p \in P} \sum_{l=1}^{L} \frac{1}{2} n_l (n_l + 1)$ correlations for any image D. The preprocessing involves computing the class-specific minimum and maximum values for the correlations: for every class $c$, the minimum and maximum values for each of the $N_S$ correlations are computed over all training examples D classified as $c$. We keep track of the minimum and maximum values of the $N_S$ correlations for all the classes in 4-D arrays Mins and Maxs, each of the order $\left( C \times L \times |P| \times \max_{1 \le l \le L} \frac{n_l(n_l+1)}{2} \right)$. Since each layer has different number of channels, the 4-th dimension has been large enough to accommodate the layer with the highest number of channels.

---

**Algorithm 1** Compute the minimum and maximum values of feature co-occurrences for each class, layer and order

---

**Input:**
    C: Number of output classes
    L: Number of Layers in entire network
    P: Set of all orders of Gram Matrix to consider
    *Train*: The train data
**Output:**
    Mins, Maxs

1: $\text{Mins}[C][L][P]\left[ \max_{1 \le l \le L} \frac{n_l(n_l+1)}{2} \right] \leftarrow \infty$       ▷ Stores the Mins for each class, layer and order

2: $\text{Maxs}[C][L][P]\left[ \max_{1 \le l \le L} \frac{n_l(n_l+1)}{2} \right] \leftarrow -\infty$       ▷ Stores the Maxs for each class, layer and order

3: **for** $c$ in $[1, C]$ **do**
4:      $Train_c = \{D \mid D \in Train \text{ s.t. } f(D) = c\}$       ▷ All the training examples predicted as $c$
5:      **for** D $\in Train_c$ **do**
6:          **for** $l$ in $[1, L]$ **do**
7:              **for** $p$ in P **do**
8:                  stat $= \overline{G_l^p}(D)$       ▷ The flattened upper triangular matrix
9:                  **for** $i$ in $\left[1, \frac{n_l(n_l+1)}{2}\right]$ **do**
10:                      $\text{Mins}[c][l][p][i] = \min(\text{Mins}[c][l][p][i], \text{stat}[i])$
11:                      $\text{Maxs}[c][l][p][i] = \max(\text{Maxs}[c][l][p][i], \text{stat}[i])$
12: **return** Mins, Maxs

---

[2]The maximum activation values observed in the convolution layers of a ResNet trained on Cifar-10 (open-sourced by Lee et al. (2018b)) are 6.5 and 6.3 on train and test partitions.

**Computing Layerwise Deviations**   Given the class-specific minimum and maximum values of the $N_S$ feature correlations, we can compute the deviation of the test sample from the images seen at train time with respect to each of the layers. In order to account for the scale of values, we compute the deviation as the percentage change with respect to the maximum or minimum values of feature co-occurrences; the deviation of an observed correlation value from the minimum and maximum correlation values observed during train time can be computed as:

$$\delta(\text{min,max,value}) = \begin{cases} 0 & \text{if min } \leq \text{value} \leq \text{max} \\ \frac{\text{min} - \text{value}}{|\text{min}|} & \text{if value} < \text{min} \\ \frac{\text{value} - \text{max}}{|\text{max}|} & \text{if value} > \text{max} \end{cases} \tag{3}$$

The deviation of a test image with respect to a given layer $l$ is the sum total of the deviations with respect to each of the $\sum_{p \in P} \frac{1}{2} n_l(n_l + 1)$ correlation values:

$$\delta_l(D) = \sum_{p=1}^{P} \sum_{i=1}^{\frac{1}{2} n_l(n_l+1)} \delta\left(\text{Mins}[D_c][l][p][i], \text{Maxs}[D_c][l][p][i], \overline{G_l^p(D)}[i]\right) \tag{4}$$

**Total Deviation** of a test image D ($\Delta(D)$), is computed by taking the sum total of the layerwise deviations ($\delta_l(D)$). However, the scale of layerwise deviations ($\delta_l$) varies with each layer depending on the number of channels in the layer, number of pixels per channel and semantic information contained in the layer. Therefore, we normalize the deviations by dividing it by $\mathbb{E}_{\text{Va}}[\delta_l]$, the expected deviation at layer $\delta_l$, computed using the validation data. Note that we use the same normalizing factor irrespective of the class assigned.

$$\Delta(D) = \sum_{l=1}^{L} \frac{\delta_l(D)}{\mathbb{E}_{\text{Va}}[\delta_l]} \tag{5}$$

**Threshold**   As is standard (Lee et al., 2018b), a threshold, $\tau$, for discriminating between out-of-distribution data and in-distribution data is computed as the 95th percentile of the total deviations of test data ($\Delta(D)$). In other words, the threshold is computed so that 95% of test examples have deviations lesser than the threshold $\tau$; the threshold-based discriminator can be formally written as:

$$\text{isOOD}(D) = \begin{cases} \text{True} & \text{if } \Delta(D) > \tau, \\ \text{False} & \text{if } \Delta(D) \leq \tau \end{cases} \tag{6}$$

**Computational Complexity.**   In order to reduce computational time, we can in fact compute deviations based on row-wise sums rather than individual elements. This would mean that the variable *stat*, defined in line 8 of Algorithm 1, would now contain row-wise sums of $G_l^p$ instead of the flattened upper triangular matrix; the inner loop of Eq. 4 would loop over $n_l$ elements instead of $\frac{1}{2} n_l(n_l + 1)$ elements while also reducing the storage required for Mins and Maxs. In practise, we found that computing the anomalies this way yields differences of less than $0.5\%$, and usually imperceptible, so the results described in the next section were computed in this way.

## 4   EXPERIMENTS - DETECTING OOD

In this section, we demonstrate the effectiveness of the proposed metric using competitive deep convolutional neural network architectures such as DenseNet and ResNet on various computer vision benchmark datasets such as: CIFAR-10, CIFAR-100, SVHN, TinyImageNet, LSUN and iSUN.

For fair comparison and to aid reproducibility, we use the pretrained ResNet (He et al., 2016) and DenseNet (Huang et al., 2017) models open-sourced by Lee et al. (2018b), i.e. ResNet34 and DenseNet3 models trained on CIFAR-10, CIFAR-100 and SVHN datasets. For each of these models, we considered the corresponding test partitions as the in-distribution (positive) examples. For CIFAR-10 and CIFAR-100, we considered the out-of-distribution datasets used by Lee et al. (2018b): TinyImagenet, LSUN and SVHN. Additionally, we also considered the iSUN dataset. For ResNet and DenseNet models trained on SVHN, we used considered CIFAR-10 dataset as the third OOD dataset. Details on these datasets are available in Appendix B.

| | Can work with pre-trained Net? | Can work without knowledge of OOD test examples? |
|---|---|---|
| DPN (Malinin and Gales, 2018) | ✗ | ✓ |
| Semantic (Shalev et al., 2018) | ✗ | ✓ |
| Variational Dirichlet (Chen et al., 2019) | ✗ | ✓ |
| Mahalanobis (Lee et al., 2018b) | ✓ | ✗ |
| ODIN (Liang et al., 2018) | ✓ | ✗ |
| OE (Hendrycks et al., 2019a) | ✗ | ✓ |
| Baseline (Hendrycks and Gimpel, 2017) | ✓ | ✓ |
| Ours | ✓ | ✓ |

Table 1: List of closely related methods. Note: OE uses OOD examples during training, but unrelated to test

We benchmark our algorithm with the works listed in Table 1 using the following metrics:

1. **TNR@95TPR** is the probability that an OOD (negative) example is correctly identified when the true positive rate (TPR) is as high as 95%. TPR can be computed as $TPR = TP/(TP + FN)$, where TP and FN denote True Positive and False Negative respectively.

2. **Detection Accuracy** measures the maximum possible classification accuracy over all possible thresholds in discriminating between in-distribution and out-of-distribution examples. For those methods which assign a higher-score to the in-distribution examples, it can be calculated as $\max_\tau \{0.5P_{in}(f(x) \geq \tau) + 0.5P_{out}(f(x) < \tau)\}$; for those methods which assign a lower score to in-distribution examples, it can be calculated as $\max_\tau \{0.5P_{in}(f(x) \leq \tau) + 0.5P_{out}(f(x) > \tau)\}$.

3. **AUROC** is the measure of the area under the plot of TPR vs FPR. For example, for those methods which assign a higher score to the in-distribution examples, this measures the probability that an OOD example is assigned a lower score than an in-distribution example.

**Experimental setup:**  We use a pre-trained network to extract class-specific minimum and maximum correlation values for all pairs of features across all orders of gram matrices. Subsequently, the total deviation is computed for each example following Eq. 5. Since the total deviation values depend on the randomly selected validation examples, we repeat the experiment 10 times to get a reliable estimate of the performance. The OOD detection performance for several combinations of model architecture, in-distribution dataset and out-of-distribution dataset are shown in Table 2. The results for Outlier Exposure (OE) are available in Table 3; some more results for OE and the results for DPN, Variational Dirichlet and Semantic are available in Appendix E.1 and Appendix E.2 respectively.

The results of Table 2 show that at a glance, over a total of 24 combinations of model architecture/in-distribution-dataset/out-of-distribution-datasets, the proposed method outperforms the previous competing methods in 15 of them, is on par in 6 of them, and gives second highest results on 3 of them[3]. Furthermore, it does so *without requiring access to samples from the OOD dataset.* If the hyperparameters and/or parameters of *Mahalanobis* and *ODIN* algorithms are fine-tuned using FGSM adversarial examples instead of the real OOD examples, their performance decreases. We also observe that our performance is similar for both architectures.

We also performed experiments with fully-connected networks by using three different MLP architectures trained on MNIST; Fashion-MNIST (Xiao et al., 2017) and KMNIST (Clanuwat et al., 2018) were considered as the out-of-distribution datasets (Results are provided in Appendix E.3).

---

[3]This is based on the TNR at TPR 95% value; AUROC and Detection Accuracy results are comparable.

| In-dist (model) | OOD | TNR at TPR 95% | AUROC | Detection Acc. |
|---|---|---|---|---|
| | | Baseline / ODIN / Mahalanobis / Ours | | |
| CIFAR-10 (ResNet) | iSUN | 44.6 / 73.2 / 97.8 / **99.3** | 91.0 / 94.0 / 99.5 / **99.7** | 85.0 / 86.5 / 96.7 / **98.1** |
| | LSUN | 49.8 / 82.1 / 98.8 / **99.6** | 91.0 / 94.1 / 99.7 / **99.8** | 85.3 / 86.7 / 97.7 / **98.6** |
| | TinyImgNet | 41.0 / 67.9 / 97.1 / **98.7** | 91.0 / 94.0 / 99.5 / **99.6** | 85.1 / 86.5 / 96.3 / **97.8** |
| | SVHN | 50.5 / 70.3 / 87.8 / **97.6** | 89.9 / 96.7 / 99.1 / **99.4** | 85.1 / 91.1 / 95.8 / **96.7** |
| CIFAR-100 (ResNet) | iSUN | 16.9 / 45.2 / 89.9 / **95.1** | 75.8 / 85.5 / 97.9 / **98.9** | 70.1 / 78.5 / 93.1 / **95.2** |
| | LSUN | 18.8 / 23.2 / 90.9 / **97.0** | 75.8 / 85.6 / 98.2 / **99.3** | 69.9 / 78.3 / 93.5 / **96.2** |
| | TinyImgNet | 20.4 / 36.1 / 90.9 / **95.1** | 77.2 / 87.6 / 98.2 / **99.0** | 70.8 / 80.1 / 93.3 / **95.1** |
| | SVHN | 20.3 / 62.7 / **91.9** / 81.4 | 79.5 / 93.9 / **98.4** / 96.2 | 73.2 / 88.0 / **93.7** / 89.8 |
| CIFAR-10 (DenseNet) | iSUN | 62.5 / 93.2 / 95.3 / **99.1** | 94.7 / 98.7 / 98.9 / **99.8** | 89.2 / 94.3 / 95.2 / **98.0** |
| | LSUN | 66.6 / 96.2 / 97.2 / **99.5** | 95.4 / 99.2 / 99.3 / **99.9** | 90.3 / 95.7 / 96.3 / **97.9** |
| | TinyImgNet | 58.9 / 92.4 / 95.0 / **98.8** | 94.1 / 98.5 / 98.8 / **99.7** | 88.5 / 93.9 / 95.0 / **97.9** |
| | SVHN | 40.2 / 86.2 / 90.8 / **96.0** | 89.9 / 95.5 / 98.1 / **99.1** | 83.2 / 91.4 / 93.9 / **95.8** |
| CIFAR-100 (DenseNet) | iSUN | 14.9 / 37.4 / 87.0 / **95.9** | 69.5 / 84.5 / 97.4 / **99.1** | 63.8 / 76.4 / 92.4 / **95.7** |
| | LSUN | 17.6 / 41.2 / 91.4 / **97.3** | 70.8 / 85.5 / 98.0 / **99.4** | 64.9 / 77.1 / 93.9 / **96.4** |
| | TinyImgNet | 17.6 / 42.6 / 86.6 / **95.8** | 71.7 / 85.2 / 97.4 / **99.0** | 65.7 / 77.0 / 92.2 / **95.6** |
| | SVHN | 26.7 / 70.6 / 82.5 / **89.4** | 82.7 / 93.8 / 97.2 / **97.4** | 75.6 / 86.6 / 91.5 / **92.4** |
| SVHN (DenseNet) | iSUN | 78.3 / 82.2 / **99.9** / 99.3 | 94.4 / 94.7 / **99.9** / 99.8 | 89.6 / 89.7 / **99.2** / 98.3 |
| | LSUN | 77.1 / 81.1 / **99.9** / 99.5 | 94.1 / 94.5 / **99.9** / 99.8 | 89.1 / 89.2 / **99.3** / 98.5 |
| | TinyImgNet | 79.8 / 84.1 / **99.9** / 99.1 | 94.8 / 95.1 / **99.9** / 99.7 | 90.2 / 90.4 / **98.9** / 97.9 |
| | CIFAR-10 | 69.3 / 71.7 / **96.8** / 80.2 | 91.9 / 91.4 / **98.9** / 95.5 | 86.6 / 85.8 / **95.9** / 89.0 |
| SVHN (ResNet) | iSUN | 77.1 / 79.1 / **99.7** / 99.1 | 92.2 / 91.4 / **99.8** / 99.8 | 89.7 / 89.2 / **98.3** / 98.1 |
| | LSUN | 74.3 / 77.3 / **99.9** / 99.4 | 91.6 / 89.4 / **99.9** / 99.8 | 89.0 / 87.2 / **99.5** / 98.5 |
| | TinyImgNet | 79.0 / 82.0 / **99.9** / 99.3 | 93.5 / 92.0 / **99.9** / 99.7 | 90.4 / 89.4 / **99.1** / 97.9 |
| | CIFAR-10 | 78.3 / 79.8 / **98.4** / 85.7 | 92.9 / 92.1 / **99.3** / 97.3 | 90.0 / 89.4 / **96.9** / 91.9 |

Table 2: Comparison of OOD Detection Performance for all combinations of model architecture and training dataset are shown. The hyperparameters of *ODIN* and the hyperparameters and parameters of *Mahalanobis* are tuned using a random sample of the OOD dataset.

| In-dist | Mean TNR @ TPR95 | | | |
|---|---|---|---|---|
| | OE (Base) | OE | Ours (Base) | Ours |
| CIFAR-10 | 65.1 | 90.5 | 51.7625 | **98.7** |
| CIFAR-100 | 37.3 | 61.5 | 19.15 | **93.4** |
| SVHN | 93.7 | **99.9** | 76.65 | 95.2 |

Table 3: Comparison of results with OE (Hendrycks et al., 2019a). Since OE uses a different model from ours, we also report the corresponding baseline accuracy. We extract the mean TNR @ TPR95 for our technique by considering both ResNet and DenseNet models. Some more results are available in Appendix E.1.

**Near-distribution Outliers** The metric does not perform well for near-distribution outliers (for example, CIFAR10 vs CIFAR100) – the detailed results can be seen in Table 4; a recent work which performs very well for near-distribution outliers and yet does not require knowledge of OOD distribution is Hendrycks et al. (2019b).

**Source of Performance Gain** The above results are obtained when we consider all elements of gram matrix (Algorithm 1), compute deviations from extrema (Eq 3) and finally, compute the total deviation with normalized layerwise deviations (Eq 5). In order to better understand the source of performance gain, we conduct a detailed ablation study that considers alternative choices for the three steps outlined before; the alternative choices are chosen in order to answer the following questions: Q1) What if strictly diagonal elements or strictly off-diagonal elements are considered instead of complete Gram Matrix?; Q2) What happens if the deviation is computed from the mean instead of the extrema?; Q3) What happens if we do not normalize the layerwise deviations? In all, we conduct 12 experiments: 3 choices for Q1 × 2 choices for Q2 × 2 choices for Q3. As a broad summary, we find that, while there is no single rule that is unbroken by an exception, our proposed combination—i.e. using the complete Gram matrix, using the min/max metric, and using normalization as in Eq 5—is generally more robust than any of the other combinations that we tried. More details on the ablation tests and discussions are available in Appendix C.

## 5 DISCUSSION AND CONCLUSION

Beyond explicit OOD detection, this line of work may ultimately help better interpret neural networks' responses to OOD examples. With this goal in mind, and at the same time to clarify the internal mechanism of our method, we perform tests to address the following two questions:

1. **Which representations are most useful?** In order to examine the role of the depth at which we compute $G$ in detecting OODs, we construct detectors which make use of correlations derived from just one residual or dense block at a time; however, all orders of gram matrices are considered. Representative results are shown in Figure 1. For all combinations of model/in-distribution/out-of-distribution-dataset, we find that the lower level representations are much more informative in discriminating between in-distribution and out-of-distribution datasets. However, the difference in detective power depends on the in-distribution dataset considered: for example, the difference in detective power between higher-level representations and lower-level representations is bigger for Cifar-100 than for Cifar-10. More graphs are available in Appendix C.1.

2. **Which orders of gram matrices are most useful?** In order to understand which orders of gram matrices are most helpful in detecting OODs, we construct detectors which make use of only one order of gram matrix at a time; however, correlations are derived from the representations of all residual and dense blocks. Representative results are shown in Figure 2. For all combinations of model/in-distribution/out-of-distribution-dataset, we find that the higher order gram matrices are much more informative in discriminating between in-distribution and out-of-distribution datasets. Ignoring the variations at orders greater than 4, we find that the TNR @ 95TPR increases with higher orders and finally saturates. More graphs are available in Appendix C in Figure 4 to 9.

**Conclusion.** Out-of-distribution detection is a challenging and important problem. We have proposed and reported on a relatively simple OOD detection method based on pairwise feature correlations that gives new state of the art detection results without requiring access to anything other than the training data itself.

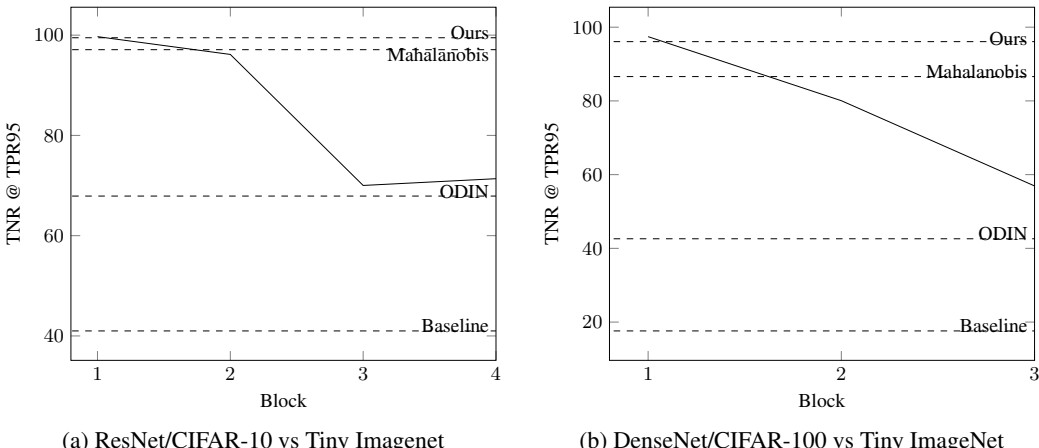

(a) ResNet/CIFAR-10 vs Tiny Imagenet      (b) DenseNet/CIFAR-100 vs Tiny ImageNet

Figure 1: Significance of depth: The TNR@TPR95 is computed by constructing detectors which make use of all the gram matrices but consider only one residual or dense block at a time. ResNet32 has 4 residual blocks and DenseNet3 has 3 dense blocks.

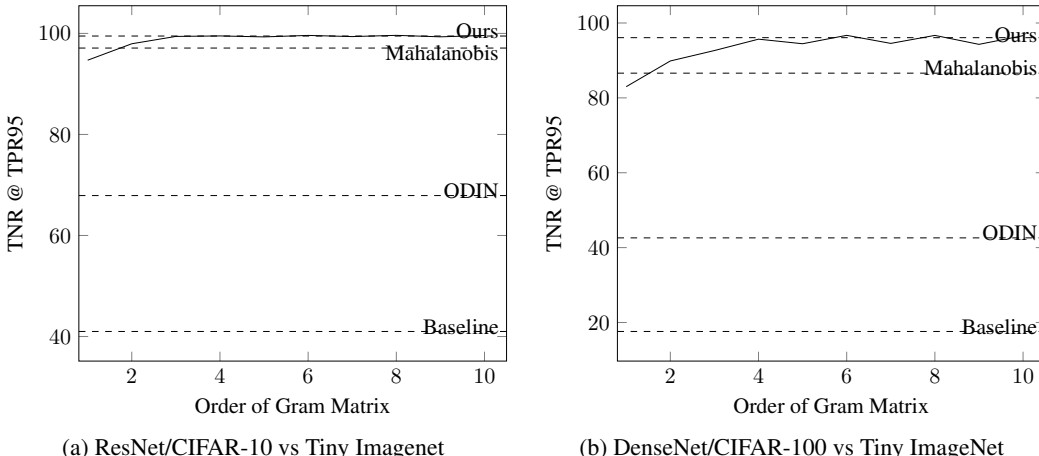

(a) ResNet/CIFAR-10 vs Tiny Imagenet

(b) DenseNet/CIFAR-100 vs Tiny ImageNet

Figure 2: The importance of higher order gram matrices: The TNR@TPR95 is computed by constructing detectors which make use of only one of the gram matrices but consider all layers.

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

# A    SCHEMATIC DIAGRAM

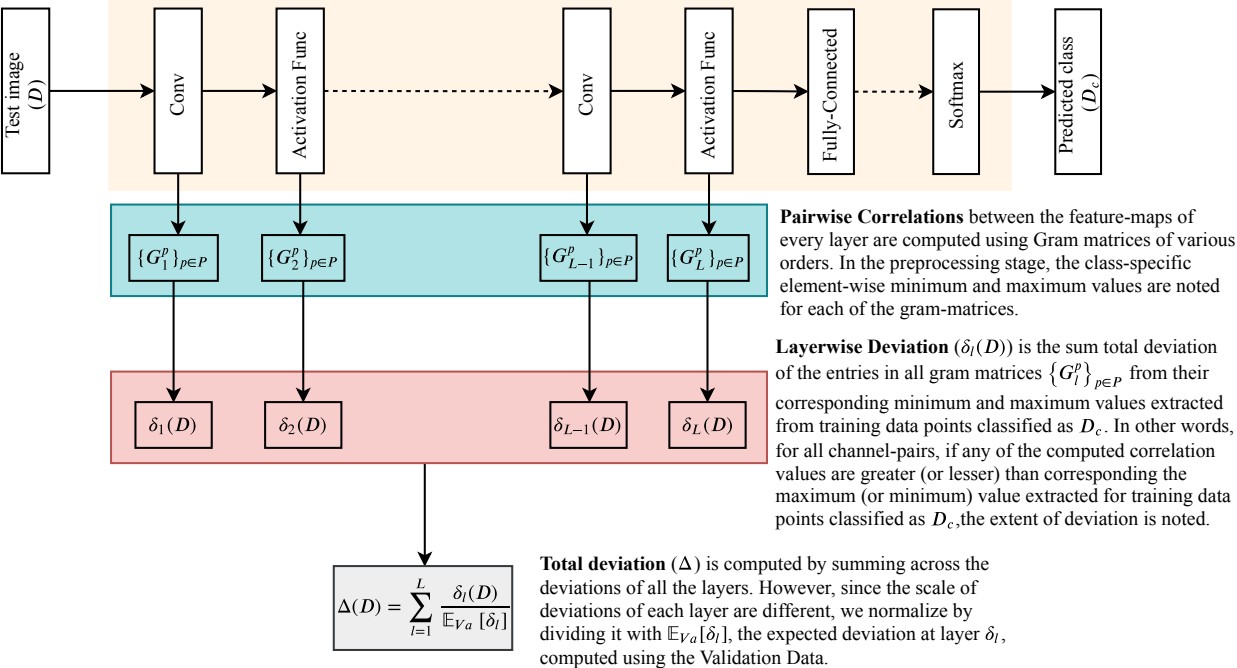

**Pairwise Correlations** between the feature-maps of every layer are computed using Gram matrices of various orders. In the preprocessing stage, the class-specific element-wise minimum and maximum values are noted for each of the gram-matrices.

**Layerwise Deviation** ($\delta_l(D)$) is the sum total deviation of the entries in all gram matrices $\left\{G_l^p\right\}_{p \in P}$ from their corresponding minimum and maximum values extracted from training data points classified as $D_c$. In other words, for all channel-pairs, if any of the computed correlation values are greater (or lesser) than corresponding the maximum (or minimum) value extracted for training data points classified as $D_c$, the extent of deviation is noted.

**Total deviation** ($\Delta$) is computed by summing across the deviations of all the layers. However, since the scale of deviations of each layer are different, we normalize by dividing it with $\mathbb{E}_{Va}[\delta_l]$, the expected deviation at layer $\delta_l$, computed using the Validation Data.

$$\Delta(D) = \sum_{l=1}^{L} \frac{\delta_l(D)}{\mathbb{E}_{Va}[\delta_l]}$$

Figure 3: The Schematic Diagram demonstrating the proposed algorithm

# B    DESCRIPTION OF OOD DATASETS

The following includes the description of the out-of-distribution datasets:

1. **TinyImagenet**, a subset of ImageNet (Russakovsky et al., 2015) images, contains 10,000 test images from 200 different classes. Each image is downsampled to size 32 x 32 and all 10,000 images are used, as given in the opensourced version by Liang et al. (2018).

2. **LSUN**, the Large-scale Scene UNderstanding dataset (Yu et al., 2015) has 10,000 test images from 10 different scenes. Each image is downsampled to size 32 x 32 and all 10,000 images are used, as given in the opensourced version by Liang et al. (2018).

3. **iSUN**, a subset of SUN images (Xiao et al., 2010), consists of 8925 images. Each image is downsampled to size 32 x 32 and is used; the downsampled version of the dataset has been opensourced by Liang et al. (2018).

4. **SVHN**, the Street View House Numbers dataset (Netzer et al., 2011), involves recognizing digits 0-9 in natural scene images. The test partition consisting of 26,032 images is used.

# C    DETAILED ABLATION RESULTS

The results in the main paper correspond to the performance obtained when considering:

1. **Feature Set**: all gram matrix entries
2. **Metric**: layerwise deviations computed with respect to the mins and maxs.
3. **Aggregation Scheme**: the total deviation is then computed using Eq 5.

In this section, detailed ablation results are reported by considering other options. Specifically:

1. **Alternate Feature Set**: In addition to considering all gram matrix entries, we consider a proper partition of the gram matrix: strictly diagonal elements, and strictly off-diagonal

elements. The diagonal elements correspond to the unary features, while the off-diagonal elements correspond to pairwise features. This can be done by appropriately changing the definition of variable *stat* in Line 7 of Algorithm 1. In these experiments, we consider row-wise sums wherever the size of stat is $O(n^2)$; in other words, we consider row-wise sums when considering off-diagonal elements and all gram matrix entries.

2. **Alternate Metric**: An alternative formulation for computing feature-wise deviations can be to compute the deviation from the means using the one-dimensional Mahalanobis distance. In the preprocessing stage, this would be done by storing the *Means* and *Variances* of *stat* (feature-wise) instead of their *Mins* and *Maxs*. Under this new alternative, the function $\delta$ defined in Eq 3 would be redefined as:

$$\delta(\text{mean,variance,value}) = \frac{(\text{value} - \text{mean})^2}{\text{variance}} \tag{7}$$

Accordingly, the layerwise deviation $\delta_l$ can be defined as:

$$\delta_l(D) = \sum_{p=1}^{P} \sum_{i=1}^{|\overline{G_l^p(D)}|} \delta\left(\text{Means}[D_c][l][p][i], \text{Variances}[D_c][l][p][i], \overline{G_l^p(D)}[i]\right) \tag{8}$$

where $\overline{G_l^p(D)}$ would correspond to the statistic chosen in the previous step: diagonal entries only, row-wise sums of off-diagonal entries or row-wise sums of complete gram matrix.

We thus consider 2 options for computing the deviations: the Min/Max method presented in the main paper and the Mean/Variance method (Gaussian) described above.

3. **Alternate Aggregation Scheme**: In order to compute the total deviation $\Delta$ from the layerwise deviations $\delta_l$, we can compute it by following 5 or taking a simple sum as shown:

$$\Delta(D) = \sum_{l=1}^{L} \delta_l \tag{9}$$

We refer to Eq 5 as the normalized estimate and Eq 9 as the unnormalized estimate.

In all, the Table 4 reports detection rates in 12 settings: 3 choices for *stat* (only the diagonal entries of gram matrix $G$, only the off-diagonal entries of $G$, or all of $G$) × 2 metrics for computing deviation (Min/Max or Mean/Variance) × 2 choices for computing total deviation (Normalized sum or Unnormalized sum). All layers and all orders of gram matrix are considered in Table 4.

| In-dist (model) | OOD | TNR at TPR95 | | | | | | | | | | | | AUROC | | | | | | | | | | | | DTACC | | | | | | | | | | | |
|---|---|---|---|---|---|---|---|---|---|---|---|---|---|---|---|---|---|---|---|---|---|---|---|---|---|---|---|---|---|---|---|---|---|---|---|---|---|
| | | Diagonal Elements | | | | Off Diagonal Elements (Row-wise Sums) | | | | Complete Gram Matrix (Row-wise Sums) | | | | Diagonal Elements | | | | Off Diagonal Elements (Row-wise Sums) | | | | Complete Gram Matrix (Row-wise Sums) | | | | Diagonal Elements | | | | Off-Diagonal Elements (Row-wise Sums) | | | | Complete Gram Matrix (Row-wise Sums) | | | |
| | | Min/Max | Mean/Var | Min/Max (U) | Mean/Var (U) | Min/Max | Mean/Var | Min/Max (U) | Mean/Var (U) | Min/Max | Mean/Var | Min/Max (U) | Mean/Var (U) | Min/Max | Mean/Var | Min/Max (U) | Mean/Var (U) | Min/Max | Mean/Var | Min/Max (U) | Mean/Var (U) | Min/Max | Mean/Var | Min/Max (U) | Mean/Var (U) | Min/Max | Mean/Var | Min/Max (U) | Mean/Var (U) | Min/Max | Mean/Var | Min/Max (U) | Mean/Var (U) | Min/Max | Mean/Var | Min/Max (U) | Mean/Var (U) |
| CIFAR-10 (ResNet) | iSUN | 99.1 | 96.3 | 95.7 | 73.8 | 99.3 | 97.1 | 97.0 | 90.2 | 99.3 | 97.1 | 97.1 | 89.3 | 99.7 | 99.2 | 98.7 | 96.2 | 99.7 | 99.4 | 99.0 | 98.1 | 99.7 | 99.4 | 99.0 | 98.0 | 97.9 | 95.7 | 95.5 | 92.1 | 98.0 | 96.2 | 96.0 | 93.5 | 98.1 | 96.2 | 96.0 | 93.4 |
| | LSUN | 99.5 | 98.3 | 97.0 | 77.7 | 99.6 | 98.8 | 98.1 | 93.6 | 99.6 | 98.8 | 98.0 | 94.1 | 99.8 | 99.5 | 98.7 | 96.7 | 99.8 | 99.7 | 99.1 | 98.5 | 99.8 | 99.7 | 99.1 | 98.4 | 98.5 | 97.0 | 96.0 | 93.0 | 98.6 | 97.6 | 96.6 | 94.6 | 98.6 | 97.6 | 96.6 | 94.5 |
| | TinyImgNet | 98.6 | 93.8 | 96.2 | 68.2 | 98.8 | 97.2 | 95.7 | 88.5 | 98.7 | 97.2 | 95.7 | 87.5 | 99.6 | 99.1 | 98.3 | 95.6 | 99.6 | 99.4 | 98.8 | 97.8 | 99.6 | 99.4 | 98.8 | 97.7 | 97.5 | 95.6 | 94.7 | 91.2 | 97.7 | 96.3 | 95.3 | 93.1 | 97.8 | 96.4 | 95.4 | 92.9 |
| | SVHN | 97.8 | 70.7 | 94.8 | 19.9 | 97.6 | 81.1 | 94.8 | 40.7 | 97.6 | 80.2 | 94.9 | 38.2 | 99.5 | 95.2 | 98.9 | 88.4 | 99.4 | 96.2 | 98.8 | 92.0 | 99.4 | 96.2 | 98.9 | 91.8 | 97.0 | 90.1 | 95.0 | 84.7 | 96.6 | 90.9 | 94.9 | 87.0 | 96.7 | 90.8 | 95.1 | 86.7 |
| | CIFAR-100 | 33.3 | 29.4 | 42.5 | 32.9 | 32.9 | 27.2 | 41.8 | 29.3 | 32.8 | 27.4 | 42.2 | 29.2 | 79.7 | 78.4 | 84.9 | 83.3 | 78.8 | 75.8 | 84.1 | 79.0 | 79.0 | 76.1 | 84.2 | 79.2 | 72.4 | 71.7 | 78.2 | 76.8 | 71.5 | 69.1 | 77.4 | 72.1 | 71.7 | 69.4 | 77.4 | 72.2 |
| CIFAR-100 (ResNet) | iSUN | 93.8 | 71.8 | 50.0 | 33.8 | 95.4 | 85.9 | 67.2 | 55.0 | 95.1 | 85.3 | 65.4 | 52.8 | 98.7 | 95.5 | 92.1 | 87.8 | 98.9 | 97.4 | 94.5 | 92.7 | 98.8 | 97.3 | 94.2 | 92.3 | 94.5 | 89.3 | 85.4 | 81.1 | 95.3 | 92.2 | 87.9 | 86.1 | 95.1 | 92.0 | 87.6 | 85.6 |
| | LSUN | 95.6 | 70.8 | 45.6 | 32.9 | 97.2 | 87.5 | 64.2 | 52.0 | 97.0 | 86.8 | 62.4 | 49.7 | 99.1 | 95.9 | 91.8 | 87.8 | 99.3 | 97.8 | 94.6 | 92.9 | 99.2 | 97.7 | 94.3 | 92.4 | 95.4 | 90.1 | 85.7 | 81.4 | 96.3 | 93.1 | 88.4 | 86.6 | 96.1 | 93.0 | 88.1 | 86.1 |
| | TinyImgNet | 94.1 | 68.0 | 51.4 | 34.6 | 95.3 | 84.2 | 68.1 | 52.8 | 95.1 | 83.5 | 66.6 | 50.8 | 98.8 | 95.1 | 92.5 | 87.1 | 99.0 | 97.2 | 94.8 | 92.5 | 98.9 | 97.1 | 94.6 | 92.1 | 94.6 | 88.5 | 85.9 | 80.1 | 95.2 | 91.8 | 88.4 | 85.9 | 95.1 | 91.6 | 88.2 | 85.4 |
| | SVHN | 83.1 | 29.8 | 53.2 | 26.0 | 79.1 | 34.4 | 51.8 | 29.6 | 81.4 | 33.9 | 55.6 | 29.2 | 96.5 | 84.7 | 92.3 | 80.8 | 95.7 | 86.8 | 91.6 | 83.5 | 96.1 | 86.7 | 92.2 | 83.1 | 90.2 | 77.5 | 85.1 | 73.6 | 89.2 | 80.3 | 84.3 | 75.3 | 89.7 | 80.3 | 84.8 | 74.9 |
| | CIFAR-10 | 12.9 | 18.1 | 19.2 | 18.2 | 11.4 | 17.5 | 17.5 | 17.9 | 12.1 | 17.6 | 18.1 | 18.0 | 69.3 | 71.2 | 76.6 | 74.6 | 67.3 | 70.0 | 75.3 | 71.9 | 67.8 | 70.1 | 75.5 | 72.0 | 64.6 | 66.0 | 71.1 | 69.1 | 63.3 | 64.8 | 69.8 | 66.5 | 63.3 | 65.0 | 70.1 | 66.6 |
| CIFAR-10 (DenseNet) | iSUN | 98.9 | 97.1 | 98.8 | 96.3 | 99.1 | 97.8 | 99.1 | 97.5 | 99.1 | 97.8 | 99.0 | 97.5 | 99.8 | 99.5 | 99.8 | 99.3 | 99.8 | 99.6 | 99.8 | 99.6 | 99.8 | 99.6 | 99.8 | 99.5 | 97.8 | 96.5 | 97.8 | 95.9 | 97.9 | 96.9 | 97.8 | 96.8 | 98.0 | 96.8 | 97.8 | 96.7 |
| | LSUN | 99.4 | 98.8 | 99.4 | 98.4 | 99.5 | 99.2 | 99.5 | 99.0 | 99.5 | 99.1 | 99.4 | 98.9 | 99.9 | 99.8 | 99.9 | 99.7 | 99.9 | 99.8 | 99.9 | 99.8 | 99.9 | 99.8 | 99.9 | 99.8 | 98.7 | 98.0 | 98.6 | 97.4 | 98.6 | 98.2 | 98.5 | 98.1 | 98.7 | 98.1 | 98.5 | 98.1 |
| | TinyImgNet | 98.7 | 97.5 | 98.6 | 97.0 | 98.8 | 98.0 | 98.8 | 97.7 | 98.8 | 97.9 | 98.7 | 97.7 | 99.7 | 99.5 | 99.7 | 99.3 | 99.7 | 99.6 | 99.7 | 99.5 | 99.7 | 99.6 | 99.6 | 99.5 | 97.9 | 96.8 | 97.8 | 96.3 | 97.8 | 97.1 | 97.7 | 97.0 | 97.9 | 97.0 | 97.7 | 97.0 |
| | SVHN | 98.6 | 87.8 | 96.9 | 88.2 | 95.9 | 84.7 | 96.4 | 86.9 | 96.0 | 84.0 | 96.5 | 87.1 | 99.2 | 97.6 | 99.3 | 97.6 | 99.1 | 96.9 | 99.2 | 97.4 | 99.1 | 96.8 | 99.2 | 97.4 | 96.3 | 92.3 | 96.5 | 92.3 | 95.8 | 91.1 | 96.0 | 92.0 | 95.8 | 90.9 | 96.1 | 92.0 |
| | CIFAR-100 | 26.4 | 28.9 | 29.0 | 28.2 | 27.0 | 25.2 | 30.1 | 24.8 | 26.8 | 25.5 | 30.1 | 25.1 | 68.5 | 75.1 | 68.9 | 74.1 | 72.1 | 72.7 | 73.3 | 72.4 | 72.2 | 72.9 | 73.4 | 72.5 | 66.5 | 68.8 | 66.6 | 67.8 | 67.6 | 67.2 | 68.9 | 66.6 | 67.3 | 67.3 | 68.6 | 66.6 |
| CIFAR-100 (DenseNet) | iSUN | 96.0 | 84.2 | 95.5 | 91.9 | 96.1 | 88.8 | 95.5 | 95.7 | 95.9 | 88.5 | 95.3 | 95.8 | 99.1 | 97.2 | 98.9 | 98.2 | 99.0 | 97.9 | 98.9 | 99.0 | 99.1 | 97.8 | 98.9 | 99.0 | 95.7 | 91.4 | 95.4 | 93.6 | 95.7 | 92.6 | 95.3 | 95.5 | 95.7 | 92.6 | 95.3 | 95.5 |
| | LSUN | 97.4 | 87.5 | 96.9 | 95.5 | 97.5 | 91.4 | 97.0 | 97.8 | 97.3 | 91.2 | 96.8 | 97.8 | 99.4 | 97.8 | 99.3 | 99.0 | 99.4 | 98.3 | 99.3 | 99.4 | 99.4 | 98.3 | 99.3 | 99.4 | 96.4 | 92.7 | 96.1 | 95.3 | 96.5 | 93.7 | 96.2 | 96.7 | 96.4 | 93.7 | 96.2 | 96.7 |
| | TinyImgNet | 95.8 | 81.4 | 95.4 | 90.2 | 95.9 | 86.9 | 95.3 | 94.2 | 95.8 | 86.4 | 95.2 | 94.3 | 99.0 | 96.6 | 98.9 | 97.8 | 99.0 | 97.5 | 98.9 | 98.7 | 99.0 | 97.4 | 98.9 | 98.7 | 95.5 | 90.4 | 95.2 | 92.9 | 95.5 | 91.8 | 95.2 | 94.7 | 95.6 | 91.7 | 95.2 | 94.7 |
| | SVHN | 89.4 | 59.7 | 88.8 | 64.5 | 87.3 | 63.2 | 86.4 | 67.4 | 89.4 | 62.9 | 87.9 | 67.3 | 97.4 | 92.5 | 97.3 | 92.6 | 97.0 | 92.7 | 96.9 | 93.4 | 97.4 | 92.7 | 97.1 | 93.4 | 92.4 | 85.7 | 92.0 | 86.0 | 91.7 | 86.0 | 91.4 | 87.1 | 92.4 | 86.2 | 91.9 | 87.1 |
| | CIFAR-10 | 10.5 | 16.4 | 11.1 | 13.7 | 10.6 | 15.6 | 10.2 | 13.9 | 10.5 | 15.6 | 10.2 | 13.8 | 64.4 | 70.1 | 65.0 | 66.7 | 63.7 | 68.3 | 64.2 | 66.2 | 64.2 | 68.7 | 64.6 | 66.2 | 60.6 | 64.9 | 61.3 | 62.2 | 59.7 | 63.4 | 60.5 | 61.6 | 60.4 | 63.8 | 61.0 | 61.5 |
| SVHN (ResNet) | iSUN | 99.3 | 99.8 | 98.1 | 96.9 | 99.5 | 99.9 | 97.8 | 98.5 | 99.5 | 99.9 | 97.9 | 98.6 | 99.8 | 99.9 | 99.8 | 98.3 | 99.8 | 99.9 | 99.0 | 99.0 | 99.8 | 99.9 | 99.0 | 99.0 | 98.1 | 98.5 | 96.4 | 96.3 | 98.1 | 98.5 | 96.4 | 96.8 | 98.1 | 98.5 | 96.5 | 96.8 |
| | LSUN | 99.6 | 99.9 | 98.5 | 97.4 | 99.6 | 99.9 | 98.3 | 99.0 | 99.6 | 99.9 | 98.4 | 99.0 | 99.9 | 99.9 | 99.8 | 98.4 | 99.8 | 99.9 | 99.1 | 99.1 | 99.8 | 99.9 | 99.1 | 99.1 | 98.5 | 98.9 | 96.8 | 96.5 | 98.5 | 98.9 | 96.7 | 97.0 | 98.5 | 98.9 | 96.7 | 97.0 |
| | TinyImgNet | 99.0 | 99.6 | 97.8 | 96.5 | 99.3 | 99.6 | 97.8 | 98.1 | 99.3 | 99.6 | 96.6 | 90.5 | 99.7 | 99.8 | 99.8 | 98.3 | 99.7 | 99.8 | 99.0 | 99.0 | 99.7 | 99.8 | 99.0 | 99.0 | 97.6 | 98.1 | 96.4 | 96.6 | 97.9 | 98.2 | 96.4 | 96.6 | 97.9 | 98.2 | 96.5 | 96.6 |
| | CIFAR-10 | 82.6 | 93.7 | 86.1 | 86.8 | 85.9 | 94.9 | 86.5 | 90.5 | 85.5 | 94.9 | 86.6 | 90.5 | 96.8 | 98.6 | 96.7 | 97.1 | 97.3 | 98.8 | 97.3 | 97.8 | 97.3 | 98.8 | 97.3 | 97.8 | 91.1 | 94.9 | 92.5 | 94.0 | 92.0 | 95.2 | 92.8 | 94.3 | 92.0 | 95.2 | 92.8 | 94.2 |
| SVHN (DenseNet) | iSUN | 99.3 | 99.6 | 99.4 | 99.1 | 99.3 | 99.4 | 99.4 | 97.9 | 99.3 | 99.4 | 99.4 | 98.0 | 99.8 | 99.9 | 99.9 | 99.8 | 99.8 | 99.9 | 99.9 | 99.5 | 99.8 | 99.9 | 99.8 | 99.5 | 98.3 | 98.8 | 98.4 | 98.3 | 98.4 | 98.5 | 98.4 | 97.3 | 98.3 | 98.6 | 98.4 | 97.3 |
| | LSUN | 99.5 | 99.7 | 99.7 | 99.4 | 99.5 | 99.4 | 99.6 | 98.2 | 99.5 | 99.4 | 99.6 | 98.3 | 99.8 | 99.9 | 99.9 | 99.9 | 99.8 | 99.9 | 99.9 | 99.6 | 99.8 | 99.9 | 99.9 | 99.6 | 98.6 | 98.9 | 98.7 | 98.5 | 98.6 | 98.6 | 98.7 | 97.6 | 98.5 | 98.7 | 98.7 | 97.7 |
| | TinyImgNet | 99.2 | 99.5 | 99.3 | 99.2 | 99.1 | 99.2 | 99.2 | 98.0 | 99.0 | 99.2 | 99.2 | 98.1 | 99.7 | 99.9 | 99.9 | 99.8 | 99.7 | 99.8 | 99.8 | 99.6 | 99.7 | 99.8 | 99.8 | 99.6 | 97.9 | 98.5 | 98.1 | 98.2 | 98.0 | 98.3 | 98.1 | 97.3 | 97.9 | 98.3 | 98.1 | 97.3 |
| | CIFAR-10 | 76.6 | 93.5 | 76.9 | 91.8 | 81.2 | 94.3 | 85.6 | 90.1 | 80.2 | 94.2 | 84.7 | 90.1 | 94.5 | 98.5 | 94.9 | 98.1 | 95.6 | 98.6 | 96.5 | 97.6 | 95.5 | 98.6 | 96.4 | 97.7 | 88.1 | 94.3 | 88.6 | 93.5 | 89.2 | 94.7 | 90.7 | 92.6 | 89.0 | 94.7 | 90.6 | 92.6 |
| Summary | MEAN | 95.4 | 86.6 | 87.9 | 77.3 | 95.6 | 90.1 | 90.4 | 83.7 | 95.7 | 89.9 | 90.4 | 83.3 | 99.0 | 97.5 | 97.7 | 95.6 | 99.0 | 98.0 | 98.1 | 96.9 | 99.0 | 98.0 | 98.1 | 96.8 | 96.0 | 93.7 | 94.1 | 91.6 | 96.1 | 94.4 | 94.5 | 92.9 | 96.2 | 94.4 | 94.5 | 92.8 |
| | STD-DEV | 6.2 | 17.2 | 18.1 | 27.0 | 6.0 | 14.7 | 13.5 | 21.3 | 5.7 | 14.9 | 13.5 | 22.0 | 1.3 | 3.4 | 2.7 | 5.2 | 1.3 | 2.9 | 2.2 | 3.9 | 1.2 | 2.9 | 2.2 | 4.0 | 2.9 | 5.2 | 4.5 | 6.8 | 2.8 | 4.5 | 3.9 | 5.6 | 2.8 | 4.5 | 3.9 | 5.7 |

Table 4: Detailed Ablation Results demonstrating the detection rates under 12 different settings. The MEAN and STD-DEV are computed by using all elements in the table excepting the CIFAR-10 vs CIFAR-100 and CIFAR-100 vs CIFAR-10 entries. *Note: We will of course fix the formatting of this for future versions; this is just an initial draft during discussion period.*

By analysing the ablation results, we attempt to answer the following questions:

1. **Are pairwise features more useful than unary features?** We observe that the Min/Max metric can work equally well with both unary and pairwise features; in some cases, the unary features are marginally better (Ex: ResNet/CIFAR-10 vs SVHN) and in some cases, the pairwise features are marginally better (Ex: ResNet/CIFAR-100 vs iSUN/LSUN/TinyImgNet). Interestingly, the behavior of the Mean/Var metric is different: the performance with pairwise features are significantly higher than with unary features in 19 out of 28 tested cases. For example, the TNR at TPR95 for ResNet/CIFAR-100 vs TinyImgNet is 68.0 with unary features and 84.2 with pairwise features.

   We notice that using the unary features (diagonal entries) sometimes did well when pairwise features (off-diagonal entries) did not do well, and vice versa, so using both gives the kind of effect that we *want* in an ensemble: models that cover and work well over different parts of the space. Therefore, an overall message of our experiments is that it is worthwhile to consider all elements of the gram matrix.

2. **Is it neccessary to use Min/Max metric?** Except in 6 cases (ResNet: CIFAR-10 vs CIFAR-100, ResNet: CIFAR-100 vs CIFAR-10, DenseNet: CIFAR-10 vs CIFAR-100, DenseNet: CIFAR-100 vs CIFAR-10, ResNet: SVHN vs CIFAR-10 and DenseNet: SVHN vs CIFAR-10), the min/max metric consistently performs better than the mean/var metric. Additionally, it is not clear if the Mean/Var estimate performs better with normalized sums or unnormalized sums: for example, observe that Mean/Var estimate performs very poorly with the unnormalized estimate for ResNet/CIFAR-100, while the performance of Mean/Var for DenseNet/CIFAR-100 is competitive with the performance of Min/Max only when an unnormalized estimate is computed.

   One can observe that computing the one-dimensional Mahalanobis distance for each component of the statistic derived from the Gram Matrix and later computing the total sum is equivalent to representing each input image by a big vector (say, $Z$) derived from the Gram Matrices computed across various layers, constructing class-conditional distributions of $Z$ (assuming that each component of $Z$ is normally distributed and independent of the other components) and subsequently computing the probability of an unseen $Z$. In early research, we noticed the following problems with the Mean/Var estimate:

   (a) The individual components of gram matrices do not follow normal distribution strictly and Mean/Var assigns lower probabilities to the in-distribution images as well.

   (b) The total deviation $\Delta$ – computed by simply summing across the layerwise deviations, $\delta_l$ – was not able to accurately summarize the information contained in the different $\delta_l$s. Specifically, information about the layer where the input example had a higher deviation was lost when a simple sum was taken.

   The proposed Min/Max idea solves problem (a) by employing a weaker metric: deviation from extrema instead of the mean. It can also be said that the Min/Max metric considers a uniform probability density between the extrema. Problem (b), which exists even for this newer metric, is solved by the normalization scheme described in the main paper for computing the sum total deviation.

**Higher Order Gram Matrices** The Min/Max metric is a weak approximation to the true probability density. On conducting a thorough analysis of how the OOD examples were able to fool the metric, it appeared that the intermediate features had several tiny activations that could yield innocuous correlation values. Higher-order gram matrices as described in the main paper provide a natural way to mitigate these effects.

Notable observations from Figures 4 through 9 (all layers are considered but only one order of gram matrix is considered at a time):

- **Ensemble effect**: In 24/28 cases, higher order gram matrices improve detection rates. Higher order gram matrices help both the Min/Max and the Mean/Var metrics. In most cases, the even powers are more helpful than the odd powers; in some cases, the odd powers are more helpful (Ex: DenseNet/CIFAR-100 vs CIFAR-10). Despite these variations, it is possible to get an ensemble effect by considering all possible powers as demonstrated in the main paper.

- In ResNet:CIFAR-10 vs CIFAR-100 and DenseNet:CIFAR-10 vs CIFAR-100, the higher order gram matrices yield lower detection rates. We find these exceptions interesting, and would like to understand them better in future.

**Summary** The unambiguous message from this ablation study is that the Gram matrix contains useful information which can be used for detecting OOD examples. While the standard Mean/Variance metric does not always work well, the proposed Min/Max metric yields consistent performance competitive with state-of-the-art methods. The use of higher-order Gram matrices further boosts the overall performance. Although the Min/Max method can work very well for "far-from-distribution" examples, it does not work well when a fine grained estimate is needed (for example, CIFAR-10 vs CIFAR-100). We hope the strong empirical proof that Gram matrices contain useful information can motivate the development of OOD detectors with powerful density estimators; eventually, such estimators might not need higher-order Gram matrices to boost their performance. Here they allow us to propose a metric based on simple thresholding that, generally, works independently of the particular OOD examples and independently of the architecture, and we believe that the fact that this is the case—the fact that this approach scores relatively well—itself raises several interesting new questions. We hope that answering these questions will ultimately strengthen our understanding of deep neural networks and give rise to more robust neural network models in the future.

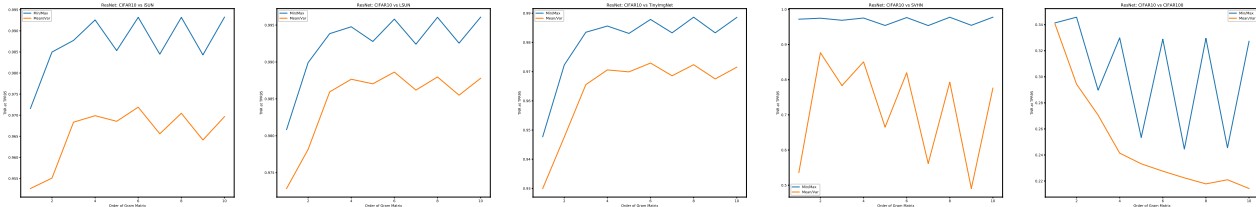

Figure 4: ResNet/CIFAR-10: The TNR at TPR95 trends for Min/Max and Mean/Var as the order of Gram Matrix is varied.

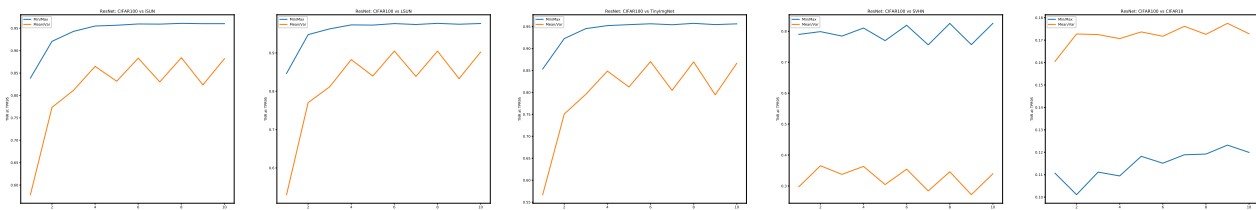

Figure 5: ResNet/CIFAR-100: The TNR at TPR95 trends for Min/Max and Mean/Var as the order of Gram Matrix is varied.

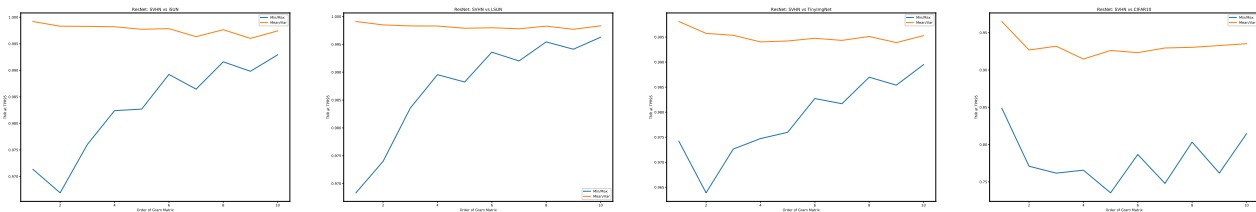

Figure 6: ResNet/SVHN: The TNR at TPR95 trends for Min/Max and Mean/Var as the order of Gram Matrix is varied.

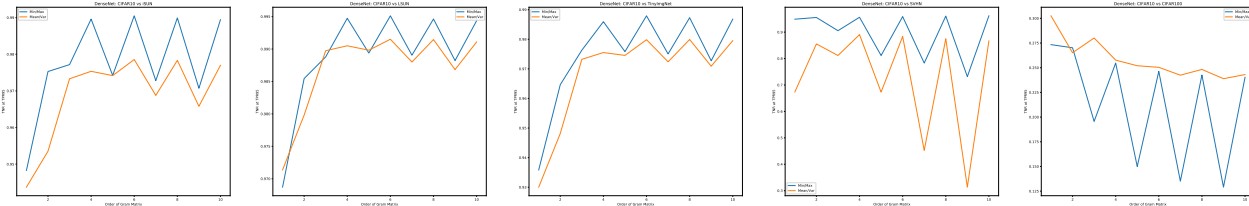

Figure 7: DenseNet/CIFAR-10: The TNR at TPR95 trends for Min/Max and Mean/Var as the order of Gram Matrix is varied.

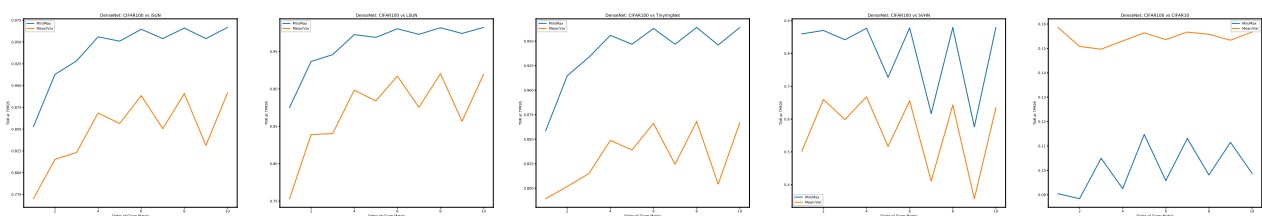

Figure 8: DenseNet/CIFAR-100: The TNR at TPR95 trends for Min/Max and Mean/Var as the order of Gram Matrix is varied.

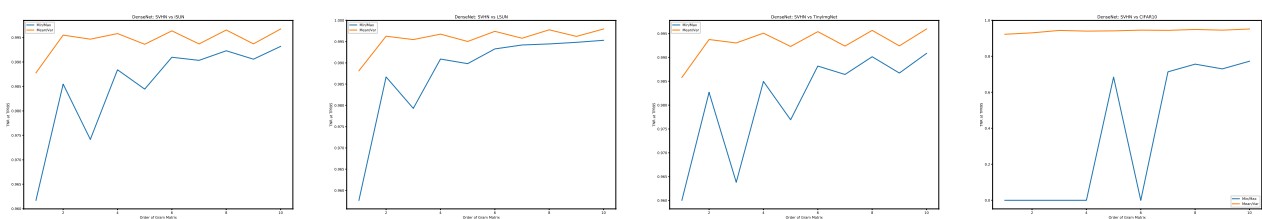

Figure 9: DenseNet/SVHN: The TNR at TPR95 trends for Min/Max and Mean/Var as the order of Gram Matrix is varied.

## C.1 SIGNIFICANCE OF DEPTH

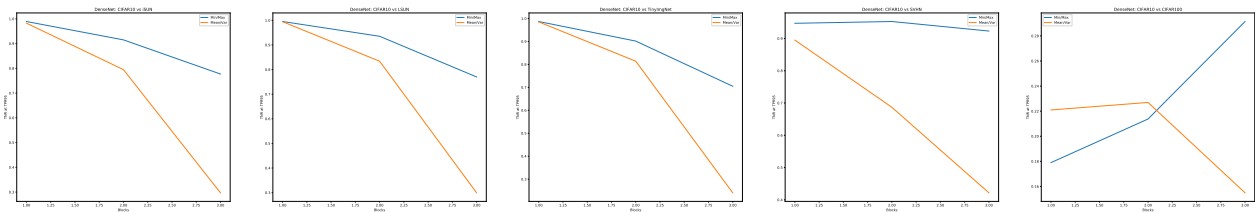

Figure 10: DenseNet/CIFAR-10: The TNR at TPR95 trends for Min/Max and Mean/Var as we go deeper in the network.

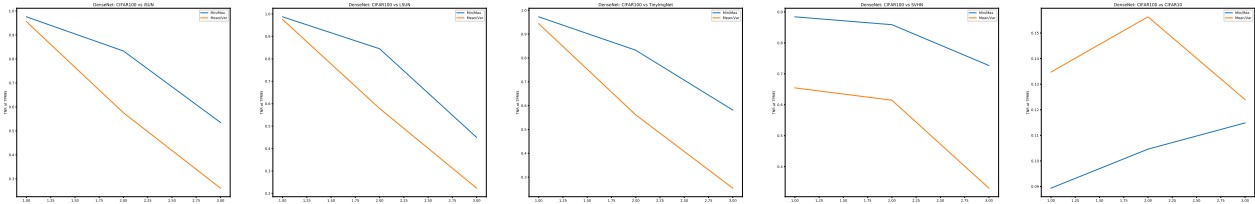

Figure 11: DenseNet/CIFAR-100: The TNR at TPR95 trends for Min/Max and Mean/Var as we go deeper in the network.

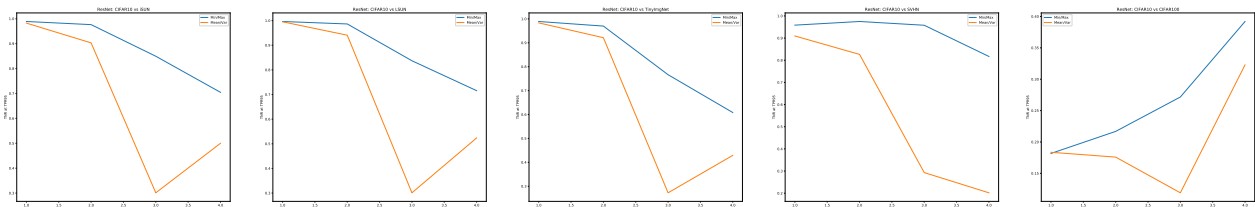

Figure 12: ResNet/CIFAR-10: The TNR at TPR95 trends for Min/Max and Mean/Var as we go deeper in the network.

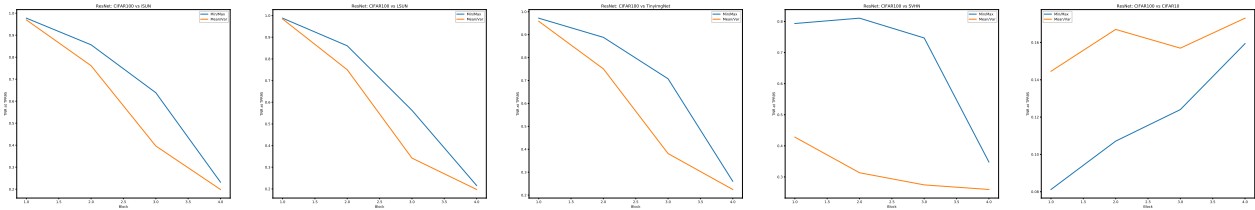

Figure 13: ResNet/CIFAR-100: The TNR at TPR95 trends for Min/Max and Mean/Var as we go deeper in the network.

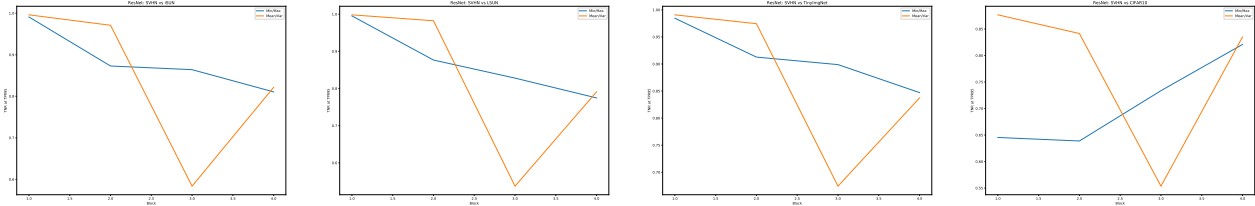

Figure 14: DenseNet/SVHN: The TNR at TPR95 trends for Min/Max and Mean/Var as we go deeper in the network.

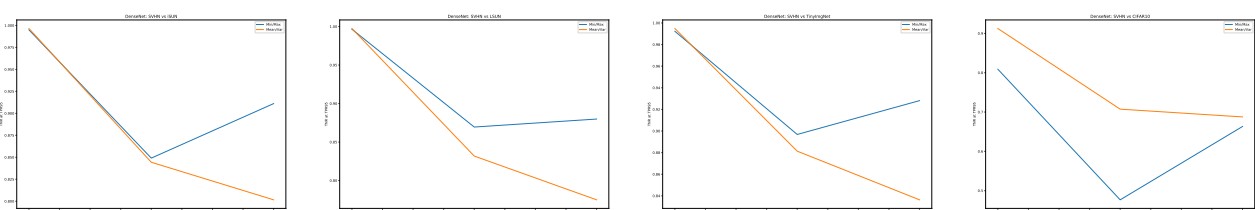

Figure 15: DenseNet/SVHN: The TNR at TPR95 trends for Min/Max and Mean/Var as we go deeper in the network.

# D  COMBINING OE + OURS

| WRN-40-2 | | Trained with Baseline | | | | | | | | | Trained with OE | | | | | | | | |
|---|---|---|---|---|---|---|---|---|---|---|---|---|---|---|---|---|---|---|---|
| | | MSP | | | Ours | | | Ours + MSP | | | MSP | | | Ours | | | Ours + MSP | | |
| In-Distribution | OOD | TNR at TPR95 | AUROC | DTACC | TNR at TPR95 | AUROC | DTACC | TNR at TPR95 | AUROC | DTACC | TNR at TPR95 | AUROC | DTACC | TNR at TPR95 | AUROC | DTACC | TNR at TPR95 | AUROC | DTACC |
| CIFAR-10 | iSUN | 43.6 | 89.9 | 83.3 | 98.7 | 99.7 | 97.5 | 98.8 | 99.7 | 97.6 | 98.3 | 99.3 | 96.9 | 98.9 | 99.8 | 97.8 | 99.8 | 100.0 | 99.1 |
| | LSUN | 47.8 | 91.5 | 85.0 | 99.3 | 99.8 | 98.3 | 99.3 | 99.8 | 98.4 | 98.5 | 99.4 | 97.0 | 99.4 | 99.9 | 98.4 | 99.9 | 100.0 | 99.3 |
| | TinyImgNet | 39.1 | 88.2 | 81.6 | 98.1 | 99.6 | 97.3 | 98.3 | 99.6 | 97.4 | 93.9 | 98.5 | 94.5 | 98.5 | 99.7 | 97.6 | 99.5 | 99.9 | 98.8 |
| | SVHN | 51.6 | 91.9 | 85.1 | 97.1 | 99.3 | 96.4 | 97.4 | 99.3 | 96.6 | 98.0 | 99.5 | 96.8 | 97.6 | 99.5 | 96.8 | 99.1 | 99.8 | 98.4 |
| | CIFAR100 | 37.0 | 87.8 | 81.5 | 25.7 | 74.7 | 68.6 | 28.3 | 75.6 | 69.2 | 73.9 | 94.8 | 87.8 | 37.8 | 80.0 | 73.2 | 58.5 | 85.1 | 78.4 |
| CIFAR-100 | iSUN | 18.4 | 78.1 | 71.6 | 96.2 | 99.1 | 95.9 | 96.2 | 99.0 | 95.8 | 50.9 | 89.8 | 82.3 | 96.3 | 99.1 | 95.9 | 96.8 | 99.3 | 96.1 |
| | LSUN | 20.2 | 79.2 | 72.5 | 97.8 | 99.4 | 96.9 | 97.8 | 99.4 | 96.8 | 58.3 | 92.0 | 84.6 | 98.4 | 99.6 | 97.2 | 98.3 | 99.6 | 97.2 |
| | TinyImgNet | 20.4 | 78.7 | 71.8 | 96.0 | 99.0 | 95.7 | 96.1 | 99.0 | 95.7 | 36.1 | 85.1 | 77.5 | 96.2 | 99.1 | 95.8 | 95.5 | 99.0 | 95.3 |
| | SVHN | 14.8 | 70.9 | 64.8 | 85.6 | 96.8 | 91.3 | 84.9 | 96.7 | 91.2 | 56.2 | 92.5 | 85.5 | 84.5 | 96.4 | 90.7 | 87.9 | 97.6 | 92.2 |
| | CIFAR10 | 16.4 | 74.0 | 67.8 | 6.0 | 58.3 | 56.3 | 8.9 | 62.8 | 59.6 | 17.4 | 78.4 | 71.7 | 7.4 | 59.1 | 57.1 | 13.8 | 69.4 | 64.9 |

Table 5: Table shows results from preliminary experiments on combining OE with our method. The experiment was conducted with pretrained WideResNet open-sourced by Hendrycks et al. (2019a). MSP uses Maximum Softmax Probability; "Ours" refers to the metric $\Delta$ (Eq. 5); "Ours+MSP" is obtained by dividing $\Delta$ with MSP.

# E  FEW MORE OOD RESULTS

## E.1  COMPARING WITH OE

| In-distribution | OOD | OE (Base) | OE | Ours (Base) | Ours |
|---|---|---|---|---|---|
| CIFAR-10 | Gaussian | 85.6 | **99.3** | 43.5 | **100.** |
| | Rademacher | 52.4 | **99.5** | 48.3 | **100.** |
| | Blob | 83.8 | **99.4** | 52.9 | **99.8** |
| | Texture | 57.2 | **87.8** | 37.0 | 85.3 |
| | SVHN | 71.2 | 95.2 | 45.4 | **96.1** |
| | LSUN | 61.3 | 87.9 | 58.2 | **99.5** |
| CIFAR-100 | Gaussian | 45.7 | 87.9 | 18.2 | **100.** |
| | Rademacher | 61.0 | 82.9 | 15.6 | **100.** |
| | Blob | 62.0 | 87.9 | 38.4 | **98.6** |
| | Texture | 28.5 | 45.6 | 19.9 | **68.5** |
| | SVHN | 30.7 | 57.1 | 23.5 | **85.4** |
| | LSUN | 26.0 | 42.5 | 18.2 | **97.2** |
| SVHN | Gaussian | 94.6 | **100.** | 87.65 | **100.** |
| | Bernoulli | 95.6 | **100.** | 92.25 | **100.** |
| | Blob | 96.3 | **100.** | 93.35 | **100.** |
| | Texture | 92.8 | **99.8** | 72.6 | 94.9 |
| | Cifar-10 | 94.0 | **99.9** | 73.8 | 83.0 |
| | LSUN | 93.6 | **99.9** | 75.7 | **99.5** |

Table 6: Comparison of Mean TNR@TPR95 values.

Following Hendrycks et al. (2019a), we created the gaussian, rademacher, blob and bernoulli synthetic datasets. Their descriptions are as follows: *Gaussian* anomalies have each dimension i.i.d. sampled from an isotropic Gaussian distribution. *Rademacher* anomalies are images where each dimension is 1 or 1 with equal probability, so each dimension is sampled from a symmetric Rademacher distribution. *Bernoulli* images have each pixel sampled from a Bernoulli distribution if the input range is [0, 1]. *Blobs* data consist of algorithmically generated amorphous shapes with definite edges. *Textures* is a dataset of describable textural images (Cimpoi et al., 2014).

### E.2 Comparing with DPN, VD and Semantic.

| OOD | Method | TNR @ TPR95 | AUROC | Detection Accuracy |
|---|---|---|---|---|
| LSUN | DPN | 42.60 | 90.20 | 79.50 |
| | VD | 92.30 | 98.30 | 94.10 |
| | Baseline | 49.80 | 91.00 | 85.30 |
| | ODIN | 82.10 | 94.10 | 86.70 |
| | Mahalanobis | 98.80 | 99.70 | 97.70 |
| | **Ours** | **99.85** | **99.89** | **98.66** |
| Tiny ImgNet | DPN | 71.60 | 93.00 | 86.40 |
| | VD | 82.90 | 96.80 | 91.30 |
| | Baseline | 41.00 | 91.00 | 85.10 |
| | ODIN | 67.90 | 94.00 | 86.50 |
| | Mahalanobis | 97.10 | 99.50 | 96.30 |
| | **Ours** | **99.48** | **99.72** | **97.82** |
| SVHN | DPN | 79.90 | 95.90 | 87.30 |
| | VD | 71.30 | 93.20 | 86.40 |
| | Baseline | 50.50 | 89.90 | 85.10 |
| | ODIN | 70.30 | 96.70 | 91.10 |
| | Mahalanobis | 87.80 | 99.10 | 95.80 |
| | **Ours** | **98.14** | **99.50** | **96.71** |

(a) ResNet/CIFAR-10

| OOD | Method | TNR @ TPR95 | AUROC | Detection Accuracy |
|---|---|---|---|---|
| iSUN | Semantic | 41.60 | 85.20 | 88.40 |
| | VD | 80.20 | 94.20 | 87.80 |
| | Baseline | 16.89 | 75.80 | 70.11 |
| | ODIN | 45.21 | 85.48 | 78.47 |
| | Mahalanobis | 89.91 | 97.91 | 93.05 |
| | **Ours** | **95.12** | **98.9** | **95.18** |
| LSUN | Semantic | 20.50 | 79.00 | 57.80 |
| | VD | 85.50 | 95.90 | 90.40 |
| | Baseline | 18.80 | 75.80 | 69.90 |
| | ODIN | 23.20 | 85.60 | 78.30 |
| | Mahalanobis | 90.89 | 98.2 | 93.5 |
| | **Ours** | **97.14** | **99.28** | **96.19** |
| Tiny ImgNet | Semantic | 37.60 | 83.10 | 75.60 |
| | VD | 83.70 | 95.30 | 89.70 |
| | Baseline | 20.40 | 77.20 | 70.80 |
| | ODIN | 36.1 | 87.6 | 80.1 |
| | Mahalanobis | 90.92 | 98.20 | 93.30 |
| | **Ours** | **95.12** | **98.97** | **95.13** |

(b) ResNet/CIFAR-100

Table 7: We compare our method with DPN, VD and Semantic by reporting results where available.

### E.3 Results for Fully-connected Networks

| Architecture | OOD | Method | TNR @ TPR95 | AUROC | Detection Accuracy |
|---|---|---|---|---|---|
| 300 | KMNIST | Baseline | 47.66 | 73.96 | 73.91 |
| | | **Ours** | **98.57** | **99.66** | **97.37** |
| | Fashion-MNIST | Baseline | 44.93 | 66.93 | 71.07 |
| | | **Ours** | **93.51** | **98.64** | **94.36** |
| 300-150 | KMNIST | Baseline | 59.79 | 75.17 | 79.49 |
| | | **Ours** | **97.8** | **99.4** | **96.55** |
| | Fashion-MNIST | Baseline | 70.73 | 77.10 | 83.00 |
| | | **Ours** | **95.2** | **99.00** | **95.17** |
| 300-150-50 | KMNIST | Baseline | 70.4 | 79.75 | 83.38 |
| | | **Ours** | **97.5** | **99.11** | **96.4** |
| | Fashion-MNIST | Baseline | 73.92 | 76.54 | 84.67 |
| | | **Ours** | **95.7** | **98.94** | **95.48** |

Table 8: The method even works quite well with a fully-connected neural network trained on MNIST. The results are shown for 300-unit single layer MLP, 300-150 two-layer MLP and 300-150-50 MLP.

