# OpenReview forum: "Zero-Shot Out-of-Distribution Detection with Feature Correlations"
_ICLR.cc/2020/Conference — Reject_

### Official Review · AnonReviewer3 · 2019-10-22
**Official Blind Review #3**

**Rating:** 8

**Review:**

This paper uses Gram matrices for OOD detection. This enables the reliable detection of far-from-distribution examples, which is a long-standing and surprisingly difficult problem in OOD detection.
This is the best paper in my batch due to the strength of the results. However, this paper should more accurately reflect the contribution: this helps with far-from-distribution examples, not near-distribution yet OOD examples.
For instance, I used their code and found that their technique leads to an AUROC of 79.01% when using CIFAR-10 as the in-distribution and CIFAR-100 as the OOD set. Likewise, with their code I found having CIFAR-100 as in and CIFAR-10 as out gives an AUROC of 67.95%. This is much worse than currently existing techniques. Hence the paper should re-frame or qualify their results as helping with far-from-distribution detection or the detection of obvious anomalies. This paper makes a solid stride in improving the detection of garbage inputs, but the paper should modify its message so as not to suggest this helps with all currently considered OOD detection tasks.

Small comments:

> Lee et al. (2018b)—to the best of our knowledge, the current SOTA technique by a significant margin
This should be again qualified and expanded. If you assume access to knowledge of the test distribution, then Mahalanobis is easily the best. If not, the Outlier Exposure is best. If you assume access to no extra data during training, then the maximum softmax probability + rotation prediction is best [1].

> However, while the OE method is able to generalize across different non-training distributions, it does not achieve the SOTA rates of Lee et al. (2018b) on most cases.
There are different senses of state-of-the-art and these should be qualified.

Does this technique do better if you do 5th and 9th percentile instead of min and max? Is it important to do the min and max with training examples instead of validation examples? (Not a pressing question.)

> Can work without access to OOD validation examples?
Table 1 is deceptive. OE does not need the "validation" examples.
I suggest two columns instead of one. "Can this work without knowledge of OOD test examples? Does this use OOD training examples?"

Show AUROC and AUPR. Detection accuracy is an unusual metric relative to AUPR (OOD as positive).

Show CIFAR-10 vs CIFAR-100 in the tables or I'll downgrade my rating, since otherwise the paper is not leaving an accurate impression.

Since OE is complementary, perhaps this technique can be combined to tackle these near-distribution cases?

The title is confusing. "Zero-Shot" could be applied to various techniques in this space. Perhaps emphasize Gram matrices?

In their code:
        validation_indices = random.sample(range(len(all_test_deviations)),int(0.1*len(all_test_deviations)))
        test_indices = sorted(list(set(range(len(all_test_deviations)))-set(validation_indices)))
These indices change with every power, which is unrealistic. Please fix the sets beforehand.

Since neural style transfer, which uses Gram matrices, works much better with VGG architectures than ResNets, does this technique work better with VGG architectures?

[1] Using Self-Supervised Learning Can Improve Model Robustness and Uncertainty. Hendrycks et al. NeurIPS 2019.

**Experience Assessment:**

I have published in this field for several years.

**Review Assessment: Checking Correctness Of Derivations And Theory:**

N/A

**Review Assessment: Checking Correctness Of Experiments:**

I carefully checked the experiments.

**Review Assessment: Thoroughness In Paper Reading:**

I read the paper thoroughly.

---

> ### Author Response · Authors · 2019-11-10
> **Response to R3.**
>
> We are grateful to R3 for their incredibly detailed review, and we are delighted that they took the time and effort to run our code. This is remarkable, and greatly appreciated. Thank you!
>
> “This paper uses Gram matrices for OOD detection. This enables the reliable detection of far-from-distribution examples, which is a long-standing and surprisingly difficult problem in OOD detection.
> This is the best paper in my batch due to the strength of the results. However, this paper should more accurately reflect the contribution: this helps with far-from-distribution examples, not near-distribution yet OOD examples.”
>
> We thank R3 for recognizing the strength of our results, and also for pointing out this important distinction regarding the contribution. We have taken initial steps to reflect this distinction, as described below, and our future revisions will do so further.
>
> “For instance, I used their code and found that their technique leads to an AUROC of 79.01% when using CIFAR-10 as the in-distribution and CIFAR-100 as the OOD set. Likewise, with their code I found having CIFAR-100 as in and CIFAR-10 as out gives an AUROC of 67.95%. This is much worse than currently existing techniques. Hence the paper should re-frame or qualify their results as helping with far-from-distribution detection or the detection of obvious anomalies. This paper makes a solid stride in improving the detection of garbage inputs, but the paper should modify its message so as not to suggest this helps with all currently considered OOD detection tasks.”
>
>
> Thank you for bringing this result to our attention. In all new ablation experiments and graphs, we have reported the results for these pairs. As of now, we have included a paragraph in the results section that notes the weakness of the metric, and places it within the context of other current results. We will obtain the results for other baseline methods and add the table as part of the main paper.
>
> Small comments:
> =========================
>
> “Lee et al. (2018b)
> [...]
> There are different senses of state-of-the-art and these should be qualified.”
>
> We agree that it is important to qualify these, and we will do so in our next revision.
>
> =========================
>
> “Does this technique do better if you do 5th and 9th percentile instead of min and max? Is it important to do the min and max with training examples instead of validation examples? (Not a pressing question.)”
>
> Sometimes, there is no change, but sometimes the result is adversely affected. If the validation examples are used to compute the min and max, there is a good chance that the test examples themselves will get a higher deviation value. In general, it is good to compute the min and max on a more representative distribution like the training examples.
>
> =========================
>
> "Can work without access to OOD validation examples? Table 1 is deceptive. OE does not need the "validation" examples. I suggest two columns instead of one. "Can this work without knowledge of OOD test examples? Does this use OOD training examples?"
>
> We have fixed this!
>
> =========================
>
> “Show AUROC and AUPR. Detection accuracy is an unusual metric relative to AUPR (OOD as positive).”
>
> We will add AUPR results for our metric and the baselines in a later revision.
>
> =========================
>
> Show CIFAR-10 vs CIFAR-100 in the tables or I'll downgrade my rating, since otherwise the paper is not leaving an accurate impression.
>
> As mentioned above, we have incorporated this into all new tables and graphs. Once we obtain results for the baselines, we will add it to Table 2.
>
> =========================
>
> Since OE is complementary, perhaps this technique can be combined to tackle these near-distribution cases?
>
> This seems to be a promising future research direction, that we would be very interested in exploring more systematically!
>
> =========================
> “In their code:
>         validation_indices = random.sample(range(len(all_test_deviations)),int(0.1*len(all_test_deviations)))
>         test_indices = sorted(list(set(range(len(all_test_deviations)))-set(validation_indices)))
> These indices change with every power, which is unrealistic. Please fix the sets beforehand.”
>
> We apologize if the code was confusing; the loop above this extracts the detection performance using 10 different random samples of the validation data. The loop over powers is done in the main ResNet/DenseNet class. We will comment the code.
>
> =========================
>
> "Since neural style transfer, which uses Gram matrices, works much better with VGG architectures than ResNets, does this technique work better with VGG architectures?"
>
> This is another very good question; we will extract the results on VGG architectures.
>
> =========================
>
> If there are any outstanding concerns that you would like us to address, we would be very happy to do so.

---

> ### Author Response · Authors · 2019-11-13
> **Small Paper Revisions**
>
> We have made several small revisions:
>
> a) Distinction of near/far-from-distribution examples in key parts (abstract, contributions, results).
>
> b) Related work section: We have replaced and rewritten a previous statement about the OE method to be more accurate: “However, while the OE method is able to generalize across different non-training distributions, it understandably does not achieve the rates of~\cite{lee2018simple}~on most cases.”
>
> c) We have also added the following in our Related Work section: “Recently, ~\cite{hendrycks2019selfsupervised} make significant advances in detecting near-distribution outliers without having any knowledge of the exact out-of-distribution examples by using in-distribution examples in a self-supervised training setting.” and another corresponding reference in our results section.
>
> We are currently working on additional small revisions, and have run some new experiments combining OE and our method, which we will describe in our next post.

---

> ### Author Response · Authors · 2019-11-13
> **Preliminary experiments on combining OE and our method**
>
> Dear R3,
>
> Inspired by your suggestion, we ran preliminary experiments on combining OE and our technique. We decided to combine the MSP (maximum softmax probability) and the total deviation $\Delta$, by dividing the $\Delta$ with the MSP. (Results are available in Table 5 on Page 17). It is possible of course that there are additional ways to combine these (we are open to suggestions).
>
> We have 2 main observations:
> a) The performance of our metric is independent of the training strategy in detecting “obvious” OODs.
> In most cases, the best results are obtained for the network trained with OE when we use “Ours + MSP”.
>
> b) When detecting near-distribution outliers, the performance of OE is better in all cases.
> When the network is trained with OE, both “Ours” and “Ours + MSP” show a considerable improvement in detecting near-distribution outliers.
> This might suggest that the richer network representations of the network trained with OE make it easier for the metric to identify outliers.
>
> At this stage, it is still hard to draw clear conclusions from this. On one hand, we suspect that the particular metric we use is not powerful enough to identify near-distribution outliers.
> On the other hand, we also suspect that the gram matrix itself might still contain useful information for identifying near-distribution outliers.
>
> An important message of this study is that: Even a normally trained neural network “can know” when an input is out-of-distribution, at least for far-from-distribution outliers.
> An exciting future research question is: How can we leverage this discovery for training neural networks which can implicitly yield confidences like OE? (We are open to any thoughts or further questions and suggestions!)
>
> Thanks again for supporting the cause of our paper!

---

### Official Review · AnonReviewer1 · 2019-10-23
**Official Blind Review #1**

**Rating:** 3

**Review:**

The paper proposes a strategy for detecting out-of-distribution samples based on feature representations obtained via neural networks. Given a test sample, the proposed strategy checks whether the correlation values between the features of the test sample obtained at different channels of the same layer are coherent with those of the training samples known to belong to the estimated class of the test sample.

The proposed strategy can be applied to pretrained networks, as it only requires the channel activation values to determine whether a sample is out of distribution or not. The studied problem is an important problem and the experimental results show that the proposed strategy leads to some performance gains in comparison to reference methods. However, in my view the main drawback of the study is that it is based on an ad-hoc methodology whose theoretical foundation is not quite clear. In particular, it would be good to provide some more explanations on the following issues:

- What exactly motivates the assumption that the correlation values between different channels of the same layer provides a discriminative characteristics of the classes from each other? It would be natural to assume that different classes will have different activation levels at a certain channel of a certain layer. However, the idea of the paper is to look at how different channels correlate with each other. I cannot entirely grasp the motivation for this, as looking at the feature correlations is a bit more indirect than looking at the features themselves. It would be good to provide the justification of this choice.

- What is the theoretical motivation behind using the p-th order Gram matrix, instead of using the original Gram matrix (e.g. p=1)? Some experimental justification is given, but it is also important to provide some theoretical insight if possible.

**Experience Assessment:**

I have read many papers in this area.

**Review Assessment: Checking Correctness Of Derivations And Theory:**

I assessed the sensibility of the derivations and theory.

**Review Assessment: Checking Correctness Of Experiments:**

I assessed the sensibility of the experiments.

**Review Assessment: Thoroughness In Paper Reading:**

I made a quick assessment of this paper.

---

> ### Author Response · Authors · 2019-11-10
> **Response to R1**
>
> We thank R1 for their careful reading of the paper and for their thoughtful and helpful questions.
>
> ====================================
> 1. "What exactly motivates the assumption that the correlation values between different channels of the same layer provides a discriminative characteristics of the classes from each other? It would be natural to assume that different classes will have different activation levels at a certain channel of a certain layer. However, the idea of the paper is to look at how different channels correlate with each other. I cannot entirely grasp the motivation for this, as looking at the feature correlations is a bit more indirect than looking at the features themselves. It would be good to provide the justification of this choice."
>
> We believe that our response to R2’s Q1, together with new ablation tests and graphs in Appendix C in the revised submission, provide justification for our method.
>
> We might also add that our work is motivated by the Mahalanobis algorithm: apart from the means, the core constituent of the Mahalanobis distance is the covariance between channels of a layer; while the Mahalanobis algorithm used a single covariance matrix independent of class and represented each channel by its mean, we decided to investigate if computing the gram matrix using all features of a channel would be useful. Roughly, the idea was to build a class-conditional distribution of gram matrices, which could later be used to compute the probability of an unseen gram-matrix at test time; also, the distribution over gram matrices would implicitly consider the distribution over the mean activations of each channel (which is essentially the Mahalanobis algorithm).
>
> ====================================
>
> 2. "What is the theoretical motivation behind using the p-th order Gram matrix, instead of using the original Gram matrix (e.g. p=1)? Some experimental justification is given, but it is also important to provide some theoretical insight if possible."
>
> As this is exactly the same question as asked by R2 Q3, please refer to our reply to that question.
>
> ====================================
>
> We believe that the additional material prompted by these questions has strengthened the overall paper. We believe we have addressed all of your concerns, but if there are any outstanding ones, we would love to have the opportunity to address them.

---

> ### Author Response · Authors · 2019-11-13
> **Further concerns after revision?**
>
> Dear Reviewer1,
>
> We found the additional experiments---prompted by your review---to be very illuminating. Now that we have completed a few other additional experiments as well, we have begun integrating the new Appendix C material into the main body of the paper for the next revision.
>
> After considering our new experiments and replies, if you have any outstanding questions or suggestions, we will be grateful to have the opportunity to address them.
>
> We thank you again for your time and insightful questions.

---

> > ### Comment · AnonReviewer1 · 2019-11-14
> > **Thanks for the response**
> >
> > I would like to thank the authors for the time they took to respond to my comments. Concerning the first comment, some additional experimental evidence is given, but some more theoretical insight would also be very useful.

---

> > > ### Author Response · Authors · 2019-11-14
> > > **Additional clarifications; Thank you for your reply!**
> > >
> > > We thank R1 for considering our new experiments and responses, and for clarifying their outstanding concerns.
> > >
> > > We agree that theoretical insights into why this technique works would help us understand and analyse deep neural networks with a new lens! Furthermore, this would also help us derive performance guarantees of the said strategy.
> > >
> > > However, this is beyond the scope of the current paper. For example, we have seen (with the help of R3) the difference in performance of our method for near versus far-from distribution examples; identifying the precise assumptions required for our method to probably work well would be in itself an interesting and non-trivial research project.
> > >
> > > In order to provide some more intuition here, we would like to point out the following:
> > > Our own suspicion is that there is, in fact, a deep connection between Gram matrices and the computations/activity representations of a deep network. For example, there is a demonstrated connection between Gram matrices and style transfer for VGG nets [1], and there is a demonstrated connection between Gram matrices and adversarial robustness for Resnets [2]. However, *exactly* what that connection is has not yet been articulated, as far as we know (beyond the algorithms themselves, of course). The present work makes a step towards this deeper connection by presenting further evidence that it exists: we do so by showing a relationship between Gram matrices and OOD detection for far-distribution examples that holds true on a variety of architectures. We did not list this as a primary contribution of the paper, and so far we did not discuss it explicitly in the paper, because it felt somewhat speculative for a research paper, especially given that our experimental results are strong enough to speak for themselves. However, we would certainly be open to any further thoughts about this---indeed it is of great interest to us---and we would be happy to include a few words about it in the paper if that seems appropriate.
> > >
> > > We claim that the paper makes a significant contribution in its current form: it empirically demonstrates that an “ordinary” pre-trained classifier can indeed reliably and consistently identify far-distribution examples without any additional information or effort. To the best of our knowledge, we are not aware of any other work which demonstrates this. We have subsequently strengthened this with ablation studies, indicating the individual and collective contribution of the elements of our method towards these results.
> > >
> > > One of the expected outcomes of this paper is to eventually propel research towards understanding “why” deep neural networks make overconfident predictions on OOD examples despite containing useful discriminatory information in their internal representations and how we might design neural networks which can “implicitly” detect OOD examples (like OE).
> > >
> > > We hope that your assessment of paper would take into account the above considerations.
> > >
> > > [1] A Neural Algorithm of Artistic Style, 2015.
> > > [2] Adversarially Robust Neural Style Transfer, Distill, 2019.

---

### Official Review · AnonReviewer2 · 2019-10-24
**Official Blind Review #2**

**Rating:** 3

**Review:**

This paper proposes a new scoring function for OOD detection based on calculating the total deviation of the pairwise feature correlations. The method only requires in-distribution data for tuning its hyper-parameters and can use a pre-trained classifier directly. Its performance is evaluated with small image datasets, which are commonly used in this line of work. Two additional experiments are performed to analyze the effect of two factors (the layer of the feature and the order of Gram matrix) in the method.

The overall vote for this paper is a weak reject. The clarity of the paper is good. The primary strength of the work is the very strong performance given the setting, which does not require OOD data for tuning. However, we give a rating slightly below the borderline, because of lack of in-depth explanation of why pairwise feature correlation is helpful for the problem. The explanation could be either theoretical or empirical, while the latter can be a set of carefully designed ablation studies. We do not see sufficient arguments or experiments to ensure the performance gain comes from the pairwise feature and is not from other parts of the design.

The above weakness could be addressed by answering the following questions:
1. If the pairwise feature (G) is replaced by the unary feature (F) while keeping the other parts of the pipeline the same (including the use of p-th power), would it reach a similar performance or much worse?
2. Why does the implementation use the statistics of min/max values instead of mean and variance? The latter can also calculate the deviation for the method using a Gaussian model, which is usually a more natural choice.
3. What is the motivation for using p-th power, and why does it help?

We present the additional feedbacks in the form of questions which could further strengthen the work if answered:
4. Based on Figure 2, having p=10 is sufficient to give a good result. Why does the method still use all p-th power (p=1..10)?
5. The study in Figure 1 only presents specific OOD data (Tiny Imagenet). Does the trend still hold with other OOD data, such as SVHN or LSUN?
6. Why are the networks used in figure 1(a) and (b) different? Does the trend still hold when the networks in both cases are changed?
7. The statistics (Mins/Maxs) to be saved can be very large. The paper provides a strategy to reduce that by using the row-wise sums of G. However, the summation has an effect of mixing the features, causing a weaker signal from the deviation. Why does the method still works with the row-wise sums?
8. Would the method work if the networks have no batch normalization layer?


**Experience Assessment:**

I have published one or two papers in this area.

**Review Assessment: Checking Correctness Of Derivations And Theory:**

I carefully checked the derivations and theory.

**Review Assessment: Checking Correctness Of Experiments:**

I carefully checked the experiments.

**Review Assessment: Thoroughness In Paper Reading:**

I read the paper thoroughly.

---

> ### Author Response · Authors · 2019-11-10
> **Response to R2 (Part 1)**
>
> We thank R2 for their careful reading, thoughtful comments, and also for phrasing their review in such a way that gives us the opportunity to answer some of these questions. We further appreciate R2’s openness to either empirical or theoretical justification. Our general approach is to provide considerably more empirical results at the moment, along with some intuition and interpretation of those results.
>
> In particular, R2 provided us with 3 focused questions to understand better where the performance gain might be coming from, and the (most welcome) suggestion that these questions might be addressed with a set of carefully designed ablation studies, and we have aimed to do exactly that. We thus compare, as described below: (1) strictly pairwise vs unary features vs both (complete Gram matrix); (2) min/max vs mean/var; and (3) with/without normalization. During our research up to this point, we have found that the latter makes a significant difference, and therefore we include it as part of this set of experiments. We have included new appendices in the revised submission that include these, which we will refer to in our detailed response below.
>
> ====================================
>
> 1. If the pairwise feature (G) is replaced by the unary feature (F) while keeping the other parts of the pipeline the same (including the use of p-th power), would it reach a similar performance or much worse?
>
> We would like to clarify that we use both diagonal elements (unary features) and off-diagonal elements (pairwise features) of the Gram Matrix. In this ablation study, we considered 2 additional settings: strictly unary features and strictly pairwise features (i.e. only the off-diagonal elements).
>
> From the detailed ablation study, we observe that the Min/Max metric can work equally well with both unary and pairwise features; in some cases, the unary features are marginally better (Ex: ResNet/CIFAR-10 vs SVHN) and in some cases, the pairwise features are marginally better (Ex: ResNet/CIFAR-100 vs iSUN/LSUN/TinyImgNet). Interestingly, the behavior of the Mean/Var metric is different: the performance with pairwise features are significantly higher than with unary features in 19 out of 28 tested cases. For example, the TNR at TPR95 for ResNet/CIFAR-100 vs TinyImgNet is 68.0 with unary features and 84.2 with pairwise features.
>
> This might suggest that the distribution of pairwise statistics are more “normal” and natural as compared to unary statistics; the pairwise statistics can be indicative (at least, partly) of unary statistics, but the converse isn’t true. Nevertheless, an overall message of our experiments is that it is worthwhile to consider all elements of the gram matrix (gives an ensemble effect).
>
> ====================================
>
> 2. Why does the implementation use the statistics of min/max values instead of mean and variance? The latter can also calculate the deviation for the method using a Gaussian model, which is usually a more natural choice.
>
> Mean/Variance was indeed the first option that we tried. In fact, computing class-conditional mean/variance for each component of the statistic derived from the Gram Matrix is equivalent to building a probability distribution over the gram matrices: Effectively, each input example can be represented by a big vector (say, $Z$) derived from the Gram Matrices computed across various layers, which can then be used to construct class-conditional distributions of $Z$ (assuming that each component of $Z$ is normally distributed and independent of the other components); this, in turn, can be used to compute the probability of an unseen $Z$. To begin with, our hypothesis was that the distribution over gram matrices can be useful; for details of our motivation, and to avoid duplication, please refer to our answer to Q1 of R1 for the motivation.
>
> In early research, we noticed the following problems with the Mean/Var estimate:
> (a) The individual components of gram matrices do not follow normal distribution strictly and Mean/Var assigns lower probabilities to the in-distribution images as well.
> (b) The total deviation $\Delta$ -- computed by simply summing across the layerwise deviations, $\delta_l$ -- was not able to accurately summarize the information contained in the different $\delta_l$s. Specifically, information about the layer where the input example had a higher deviation was lost when a simple sum was taken.
>
> The proposed Min/Max idea solves problem (a) by employing a weaker metric: deviation from extrema instead of the mean. It can also be said that the Min/Max metric considers a uniform probability density between the extrema. Problem (b), which exists even for this newer metric, is solved by the normalization scheme described in the main paper for computing the sum total deviation.
>
> We refer the reader to Appendix C for a comprehensive summary of challenges with the Mean/Var metric, and also to the computed means and variances (in the last row of Table 4).

---

> > ### Author Response · Authors · 2019-11-10
> > **Response to R2 (Part 2/2)**
> >
> > 3. What is the motivation for using p-th power, and why does it help?
> >
> > The Min/Max metric is a weak approximation to the true probability density. On conducting a thorough analysis of how the OOD examples were able to fool the metric, it appeared that the intermediate features had several tiny activations that could yield innocuous correlation values. Higher-order gram matrices as described in the main paper provide a natural way to mitigate these effects by suppressing tiny activation values.
> >
> > Thus, we have found that it is not possible to effectively test against class-conditional distributions over gram matrices with probability density estimators such as Min/Max or Mean/Var without the higher-order gram matrices.
> >
> > An essential and unambiguous message from this ablation study is that the Gram matrix contains useful information which can be used for detecting OOD examples. While the standard Mean/Variance metric does not always work well, the proposed Min/Max metric yields consistent performance competitive with state-of-the-art methods. The use of higher-order Gram matrices further boosts the overall performance.
> >
> > We hope the strong empirical proof that Gram matrices contain useful information will motivate the development of OOD detectors with powerful density estimators in the future; eventually, such estimators might not need higher-order Gram matrices to boost their performance.
> >
> > ====================================
> >
> > 4. Based on Figure 2, having p=10 is sufficient to give a good result. Why does the method still use all p-th power (p=1..10)?
> >
> > In order to answer this question accurately, we have now included all the graphs; please refer to figures 4 through 9 of the revised submission.
> >
> > There exist certain variations and p=10 alone might not be the universally ideal choice. For example, observe that p=1 gives a generally better detection rate for Min/Max in ResNet/SVHN vs CIFAR10. For ResNet/SVHN, it can also be observed that Mean/Var performs better with p=1 in all tested cases. In most cases, the even powers are more helpful than the odd powers; in some cases, the odd powers are more helpful (Ex: DenseNet/CIFAR-100 vs CIFAR-10). In 24/28 cases, higher order gram matrices do improve detection rates. Higher order gram matrices help both the Min/Max and the Mean/Var metrics.
> >
> > Overall— and this can be seen by looking at the new detailed graphs—while there are certain patterns, they do not hold in every case, and so we have found that there is an ensembling effect that occurs across the multiple powers, that contributes to the robustness of the method across datasets and architectures.
> >
> > ====================================
> >
> > 5. The study in Figure 1 only presents specific OOD data (Tiny Imagenet). Does the trend still hold with other OOD data, such as SVHN or LSUN?
> > 6. Why are the networks used in figure 1(a) and (b) different? Does the trend still hold when the networks in both cases are changed?
> >
> > We have added graphs for all cases in Appendix C.1.
> >
> > In 23/28 graphs, both Min/Max and Mean/Var follow this trend. In 3/28 graphs, Mean/Var follows this trend and Min/Max doesn’t clearly follow this trend. In 1 graph (ResNet:CIFAR-100 vs CIFAR-10), both of them do not follow this trend.
> >
> > In our experience, as with the question above, it is indeed a trend and not a rule: it seems to be the case for many of the pairs, but there are some clear exceptions.
> >
> > ====================================
> >
> > 7. The statistics (Mins/Maxs) to be saved can be very large. The paper provides a strategy to reduce that by using the row-wise sums of G. However, the summation has an effect of mixing the features, causing a weaker signal from the deviation. Why does the method still works with the row-wise sums?
> >
> > This is a good observation, and we agree that summation does mix the features. Empirically, we have found that the signal obtained using the Gram matrices is strong enough that it can withstand this operation.
> >
> > ====================================
> >
> > 8. Would the method work if the networks have no batch normalization layer?
> >
> > While we have not tried this on networks without any batch normalization at all, based on our experience (e.g. computing statistics on layers before any batch norm; ablation study using only the unary features), we do not see any indication that batch norm affects the results at all.
> >
> > ====================================
> >
> > We believe we have addressed all of your concerns, but if there are any outstanding ones, we would love to have the opportunity to address them.

---

> ### Author Response · Authors · 2019-11-13
> **Further concerns after revision?**
>
> Dear Reviewer2,
>
> We found the additional experiments---prompted by your review---to be very illuminating. Now that we have completed a few other additional experiments as well, we have begun integrating the new Appendix C material into the main body of the paper for the next revision.
>
> After considering our new experiments and replies, if you have any outstanding questions or suggestions, we will be grateful to have the opportunity to address them.
>
> We thank you again for your time and insightful questions.

---

### Author Response · Authors · 2019-11-15
**Summary of Comments and Contributions**

As the rebuttal period is closing soon, we would like to thank all of the referees once again (as well as the AC), and provide a summary of the discussion from our perspective.

The paper proposes an algorithm for detecting out-of-distribution examples using nothing more than a pretrained classifier and the train/validation partitions of the in-distribution examples.

All three reviewers agree that despite this constraint, the results are strong:
R1: “... the experimental results show that the proposed strategy leads to some performance gains in comparison to reference methods.”
R2: “The primary strength of the work is the very strong performance given the setting, which does not require OOD data for tuning”
R3: “This is the best paper in my batch due to the strength of the results.”

The method is straightforward and makes use of Gram matrices at multiple layers, with higher-order Gram matrices for better overall performance. Test-time Gram matrix entries are compared to those observed at train-time.

R3 emphasized the distinction between near and far-from-distribution examples---we appreciate this distinction---and pointed out that it is on far-from-distribution examples that our method performs exceptionally well. We subsequently ran new experiments to incorporate this into the paper (and will include new baselines when available). For detecting far-distribution outliers, the proposed strategy performs better than or equal to state-of-the-art OOD detection methods (the latter which require specific OOD samples).

This strong performance raises several important and interesting questions (as asked by R2 and R1). We summarize the two themes of these questions as:
(a) ` `  why does this work?’’ (R1 and R2) and
(b) “exactly what part of this works?” (R2 suggested we at least answer (b), if not (a).)

We thus did as R2 suggested: we conducted extensive ablation tests and incorporated the results in the paper and in (the new) Appendix C. A summary of the discussion can be found in the responses to R2 and R1. Broadly speaking, we found that our proposed method tended to be more robust than any of the other ~12 ablative combinations that we tried.

For our thoughts on question (a), see our reply to R1 entitled “Additional clarifications; Thank you for your reply!”.

We note that the question “how and why does the Gram matrix contain so much useful information about the network?” is exactly one of the questions that we hope that this paper will raise in the community, through the strength of the results and the relative simplicity of the method to obtain them.

We demonstrate that without extra information, a pre-trained classifier can indeed know what examples are in-distribution and what examples are out-of-distribution, at least for far-distribution examples.
We believe that this is a significant discovery (R3: “This paper makes a solid stride in improving the detection of garbage inputs”).

---

### Decision · Program_Chairs · 2019-12-19

**Decision:**

Reject

**Comment:**

The paper proposes a new scoring function for OOD detection based on calculating the total deviation of the pairwise feature correlations. This is an important problem that is of general interest in our community.

Reviewer 2 found the paper to be clear, provided a set of weaknesses relating to lack of explanations of performance and more careful ablations, along with a set of strategies to address them. Reviewer 1 recognized the importance of being useful for pretrained networks but also raised questions of explanation and theoretical motivations. Reviewer 3 was extremely supportive, used the authors' code to highlight the difference between far-from-distribution behaviour versus near-distribution OOD examples. The authors provided detailed responses to all points raised and provided additional eidence. There was  no convergence of the review recommendations.

The review added much more clarity to the paper and it is no a better paper. The paper demonstrates all the features of a good paper, but unfortunately didn't yet reach the level for acceptance for the next conference.